# 3D-FREE MEETS 3D PRIORS: NOVEL VIEW SYNTHESIS FROM A SINGLE IMAGE WITH PRETRAINED DIFFUSION GUIDANCE

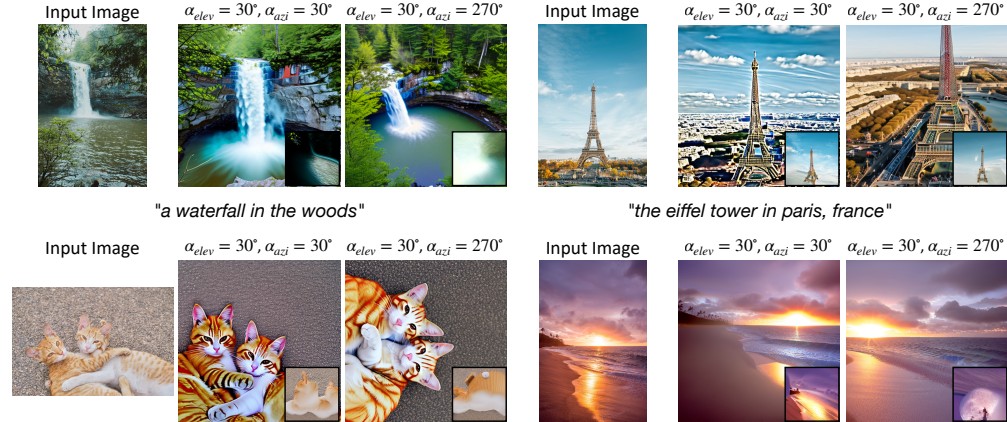

Figure 1: Our model is capable of generating high quality camera-controlled images at specific azimuth and elevation angles for a variety of complex scenes, all without requiring extra 3D datasets or extensive training. The image in the bottom right corner showcases the output from the 3D-based baseline, Zero123++ Shi et al. (2023a), created from a designated angle.

## ABSTRACT

Recent 3D novel view synthesis (NVS) methods often require extensive 3D data for training, and also typically lack generalization beyond the training distribution. Moreover, they tend to be object centric and struggle with complex and intricate scenes. Conversely, 3D-free methods can generate text-controlled views of complex, in-the-wild scenes using a pretrained stable diffusion model without the need for a large amount of 3D-based training data, but lack camera control. In this paper, we introduce a method capable of generating camera-controlled viewpoints from a single input image, by combining the benefits of 3D-free and 3D-based approaches. Our method excels in handling complex and diverse scenes without extensive training or additional 3D and multiview data. It leverages widely available pretrained NVS models for weak guidance, integrating this knowledge into a 3D-free view synthesis style approach, along with enriching the CLIP vision-language space with 3D camera angle information, to achieve the desired results. Experimental results demonstrate that our method outperforms existing models in both qualitative and quantitative evaluations, achieving high-fidelity, consistent novel view synthesis at desired camera angles across a wide variety of scenes while maintaining accurate, natural detail representation and image clarity across various viewpoints. We also support our method with a comprehensive analysis of 2D image generation models and the 3D space, providing a solid foundation and rationale for our solution.

Figure 2: **Method.** Our method generates a high fidelity camera controlled novel viewpoint of a single image $I_{input}$, its text description and designated angle information. It infuses prior information from pre-trained NVS models into the text to image stable diffusion architecture in a 3D-free inference-time optimization procedure.

# 1 INTRODUCTION

Novel-view synthesis plays a pivotal role in numerous real-world applications, including 3D environments, augmented reality (AR), virtual reality (VR), and autonomous driving. Recent advancements in diffusion-based methods, such as Zero-1-to-3 (Zero123) Liu et al. (2023a) and Zero123++ Shi et al. (2023a), along with NeRFs Tancik et al. (2022); Zhou & Tulsiani (2023); Deng et al. (2023) Gaussian Splatting Li et al. (2024); Zhu et al. (2023), and so on Sargent et al. (2023); Tung et al. (2025); Van Hoorick et al. (2025); Burgess et al. (2023); Wiles et al. (2020); Shen et al. (2021); Tucker & Snavely (2020). They have significantly propelled the field forward. Some techniques enable the specification of camera angles and the sampling of novel-view images from precise viewpoints. However, diffusion-based methods remain largely object-centric and may struggle to generalize to complex scenes with intricate backgrounds. They also require extensive 3D object datasets for training. In contrast, NeRF and Gaussian Splatting methods can handle complex scenes but depend on multi-view information to construct 3D models. Therefore, achieving novel-view synthesis from a single image in a data-efficient manner, without relying on additional 3D, multi-view, or depth information, is highly advantageous.

On the other hand, 3D-free methods such as DreamBooth and other recent models Kothandaraman et al. (2023b;a); Ruiz et al. (2023) aim to intelligently extract the rich 3D knowledge embedded in text-to-2D image diffusion models, such as stable diffusion Rombach et al. (2022); Podell et al. (2023), to generate text-controlled views from complex, real-world input images without needing additional multi-view or 3D information for fine-tuning or inference. Among these, HawkI stands out as the current best 3D-free model, demonstrating superior capability in utilizing embedded 3D knowledge for high-quality, text-controlled image synthesis. However, despite its excellence, HawkI, like other 3D-free methods, lacks the ability to precisely control camera angles when generating novel-view images. Ideally, we aim for both data-efficient novel view synthesis and camera controllability, which is the primary focus of this paper.

We start by examining why 3D-free methods like HawkI struggle with camera control. To understand this, we need to assess how effectively the CLIP model—used as the vision-language backbone in image generation models like Stable Diffusion—interprets the 3D space. Our analysis shows that while CLIP excels at recognizing scene entities and general directions (such as up, down, left, and right), it falls short in grasping specific angles, like 30 degrees upward. This limitation makes it inadequate for generating camera-controlled views on its own. Therefore, to achieve camera control, we need guidance on angles, which can be provided by 3D priors from pretrained 3D models. One approach is to integrate this 3D prior information into 3D-free models like HawkI.

Before we explore how to incorporate these 3D priors into HawkI, it's crucial to understand the role of guidance images in 3D-free methods. Our analysis indicates that incorrect guidance can lead to significant inconsistencies in the generated images, both in terms of angle and content. Thus, it's essential for the 3D prior to accurately understand angles and to be effectively utilized.

Using these insights, along with the established knowledge that 3D-based methods such as HawkI enable precise camera control and 3D-free methods like Zero123++ offer generalizability and data

efficiency, we propose a simple approach for novel-view synthesis that generates camera-controlled novel views at specified azimuth and elevation angles from a single input image, without requiring 3D datasets or extensive training. Our method utilizes information from off-the-shelf pretrained model, specifically using Zero123++ Shi et al. (2023a), a plug-and-play model, in conjunction with the pretrained stable diffusion model. The process employs a 3D-free HawkI-style optimization procedure during inference to achieve the desired outcomes, utilizing information from 3D-based methods as pseudo or weak guidance images. To improve viewpoint consistency—an area where the CLIP model lacks information—we introduce a regularization loss term. This term promotes alignment between the target angle embedding (which captures elevation and azimuth data) and the optimized embedding. By integrating 3D angular information within the CLIP space through the 3D prior image, we reinforce the specified camera viewpoint in the generated images. We validate our approach through extensive qualitative and quantitative comparisons across various metrics that assess text consistency and fidelity w.r.t. input image.

In summary, the contributions of this paper are as follows: (1) We present a novel approach for novel view synthesis that allows for precise camera control, especially effective for complex images with multiple objects and detailed backgrounds. Our method harnesses insights from pretrained 3D models within a 3D-free framework, removing the necessity for additional multi-view or 3D data during both training and inference, effectively combining the advantages of both approaches. (2) We provide an analysis of the CLIP model's understanding of 3D space and the role of guidance images in 3D-free methods. This analysis supports our solution by highlighting the importance of using priors from pretrained 3D models to enhance viewpoint information in 3D space, utilizing the 3D prior image as a guiding factor for our task. (3) We present comprehensive qualitative and quantitative results on various synthetic and real images, demonstrating significant improvements over baseline 3D-based and 3D-free methods in terms of text consistency and fidelity relative to the input images. Our model's results maintains consistent, accurate, natural detail representation and image clarity across various viewpoints. Also, our model outperforms the lowest-performing model by **0.1712 in LPIPS** (HawkI-Syn $(-20°, 210°)$ in Table 5), which is **5.2 times larger than** the 0.033 gap of Zero123++ in comparison to the lowest-performing model (Table 1 in Shi et al. (2023a)).

## 2 PRIOR WORK ON 3D AND 3D-FREE APPROACHES FOR NVS

Recent research has increasingly focused on novel view synthesis using diffusion models Chen et al. (2021); Mildenhall et al. (2021); Shi et al. (2021); Gu et al. (2023). Approaches for 3D generation Chen et al. (2024); Lin et al. (2023); Poole et al. (2022); Raj et al. (2023); Xu et al. (2023); Gao et al. (2024); Park et al. (2017) often rely on text for reconstruction and require substantial multi-view and 3D data Shi et al. (2023b); Wang & Shi (2023); Yang et al. (2024); Liu et al. (2023b); Höllein et al. (2024); Jain et al. (2021); Liu et al. (2024); Shi et al. (2023c;b) for supervised learning. For instance, Zero123 Liu et al. (2023a), Magic123 Qian et al. (2023), and Zero123++ Shi et al. (2023a) utilize a pre-trained stable diffusion model Rombach et al. (2022) combined with 3D data corresponding to 800k objects to learn various camera viewpoints. In other words, they are extremely data hungry, meaning that they need extensive multi-view and 3D data to train on. Additionally, most state-of-the-art view synthesis algorithms are largely object-centric, and may not work well on complex scenes containing multiple objects or background information. This is due to the domain gap between the 3D objects data that they are typically trained on, and the inferencing image.

On the other hand, Free3D Zheng & Vedaldi (2024) introduces an efficient method for synthesizing accurate 360-degree views from a single image without 3D representations. By incorporating the Ray Conditioning Normalization (RCN) layer into 2D image generators, it encodes the target view's pose and enhances view consistency with lightweight multi-view attention layers and noise sharing. However, it still requires training on large-scale 3D datasets like Objaverse and multi-view information, and it cannot include background transformations. DreamFusion Poole et al. (2022) presents a Text-to-3D method using a NeRF and a Diffusion Model-based Text-to-2D model. It introduces a probability density distillation loss, allowing the 2D Diffusion Model to optimize image generation without needing 3D data or model modifications. DreamFusion's key contributions are creating Text-to-3D models without 3D dataset training and utilizing a Diffusion Model. However, it cannot transform images with backgrounds or introduce elevation changes in camera-controlled images. Aerial Diffusion and HawkI Kothandaraman et al. (2023a;b) synthesize high-quality aerial view images using text and a single input image without 3D or multi-view information. They employ a

pretrained text-to-2D Stable Diffusion model, achieving a balance between aerial view consistency and input image fidelity through test-time optimization and mutual information-based inference. However, Aerial Diffusion doesn't extend well to complex scenes and has artifacts in the generated results, and HawkI struggles with controlling camera angles, detailed feature generation, and maintaining view consistency.

# 3 UNDERSTANDING 2D MODELS AND THE 3D SPACE

In this section, we use 3D-free stable diffusion based view synthesis method, HawkI Kothandaraman et al. (2023b). HawkI employs classical computer vision techniques to generate aerial view images from ground view images through a homography transformation, which acts as the guidance image for the diffusion model. We used the HawkI default setting for experiments, unless stated otherwise.

## 3.1 HOW WELL DO CLIP MODELS UNDERSTAND THE 3D-SPACE?

The main reason stable-diffusion-based 2D models using 3D-free approaches struggle with camera control is their limited understanding of 3D space. While they can grasp concepts like top, bottom, and side, they lack precise camera information, such as "30 degrees to the right." A key factor in this limitation is the CLIP model, which serves as the vision-language backbone for models like Stable Diffusion. In this section, we analyze how effectively CLIP models comprehend 3D space by examining the 3D-free stable diffusion based view synthesis method, HawkI Kothandaraman et al. (2023b). Our hypothesis is that HawkI's capability to execute viewpoint transformations without relying on 3D data is dependent on this homography image.

We conducted an experiment where the homography image was omitted to see if CLIP could generate camera-controlled images without the guidance factor for camera angle control. Detailed angle instructions were still provided in the target text description. Our results showed that CLIP could not independently generate consistent viewpoints, highlighting the importance of 3D guidance images. In the experiment, different pyramids, waterfalls, and houses were generated inconsistently, and camera control angles were not accurately followed. This demonstrates that CLIP struggles with 3D comprehension and validates the necessity of novel view image (guidance image). Figure 3 illustrates these findings.

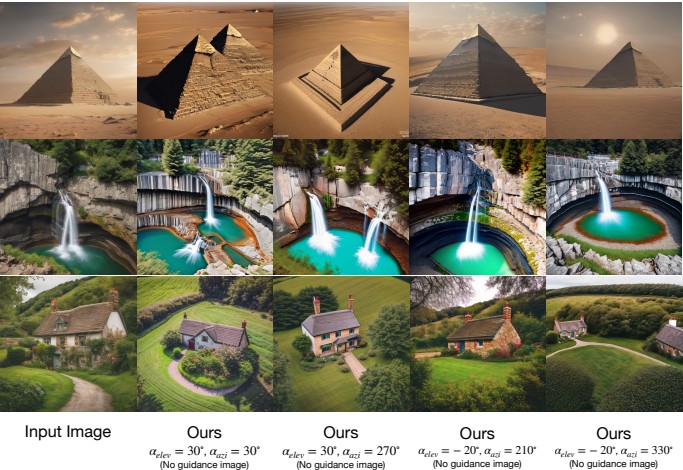

| Input Image | Ours $\alpha_{elev} = 30°, \alpha_{azi} = 30°$ (No guidance image) | Ours $\alpha_{elev} = 30°, \alpha_{azi} = 270°$ (No guidance image) | Ours $\alpha_{elev} = -20°, \alpha_{azi} = 210°$ (No guidance image) | Ours $\alpha_{elev} = -20°, \alpha_{azi} = 330°$ (No guidance image) |

Figure 3: **Analysis of how well CLIP understands the 3D space** In this experiment, we generate camera control images for specific angles without using any guidance image.

## 3.2 IMPORTANCE OF GUIDANCE IMAGE IN 3D-FREE METHODS

Our previous analysis revealed that the CLIP model in the view synthesis method (HawkI) without a guidance image struggles to understand 3D space, resulting in inconsistent images. Conversely, HawkI with a guidance image cannot perform transformations from various camera viewpoints. This raises the question of what kind of guidance image is suitable for 3D-free camera control.

We conduct experiments using the images generated using the pretrained Zero123++ Shi et al. (2023a) model for guidance. In our experiments, the target text specified the desired transformation angles, but the guidance images had different angles. i.e. the guidance images introduced incorrect viewpoints. When generating an image at a $(30°, 30°)$ angle, the model followed the guidance image's suggestion, regardless of the text input. This emphasizes that the model benefits from the information in the 3D-prior model's guidance image a lot. Our experiment highlights the importance of accurate guidance images for camera control. Figure 4 illustrates these findings.

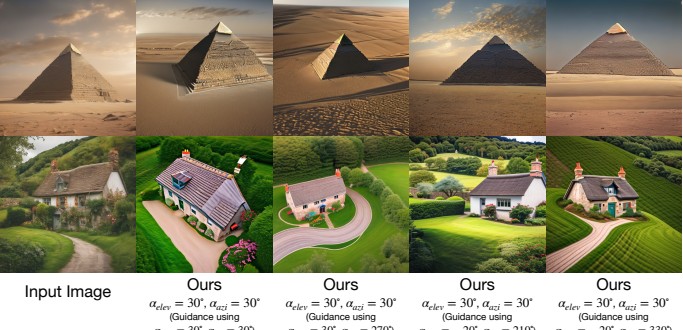

Input Image    Ours    Ours    Ours    Ours

Figure 4: **Using an image with an incorrect viewpoint as the guidance image** In this experiment, we examine how the results are derived when an incorrect viewpoint image is used as a guidance image.

# 4 3D-FREE MEETS 3D PRIORS: AN APPROACH FOR 3D-DATA EFFICIENT NVS

According to the analysis, we present a method (Figure 2) for data-efficient text and camera-controlled novel-view synthesis from a single input image ($I_{input}$) and its text description ($t_{input}$) (e.g., "An ancient Egyptian pyramid in the desert," obtained using the BLIP-2 Li et al. (2023) model). Our model eliminates the need for training data, 3D data, or multi-view data. Instead, it utilizes a pretrained text-to-2D image stable diffusion model as a strong prior, along with pretrained novel-view synthesis (NVS) models, e.g. Zero123++, for guidance. Our method combines information from the pretrained NVS model and performs a rapid inference-time optimization and inference routine to generate novel-view images of any given in-the-wild complex input scene at specified elevation ($\alpha_{elev}$) and azimuth angles ($\alpha_{azi}$). Elevation ($\alpha_{elev}$) refers to the vertical angle relative to the object, measured in degrees, and is defined based on the orientation of the input image. Similarly, azimuth angles ($\alpha_{azi}$) refer to the horizontal angle around the object, also relative to the input image. We next describe our method in detail.

## 4.1 INFERENCE-TIME OPTIMIZATION

We employ a pretrained NVS model $G$ to obtain a weak prediction, $I_{view}$, of $I_{input}$ at ($\alpha_{elev}, \alpha_{azi}$). This prediction is represented as $I_{view} = G(I_{input}, \alpha_{elev}, \alpha_{azi})$. Although $I_{view}$ is not a fully accurate depiction of the desired target, it provides weak or pseudo guidance for the model regarding the content and direction of the desired viewpoint transformation. Subsequently, we utilize the pretrained text-to-image stable diffusion model to perform inference-time optimization Kothandaraman et al. (2023b).

Across all four steps, the reconstruction loss $L$ is used to guide the optimization process, ensuring accurate reconstruction of $I_{input}$ and $I_{view}$. In Step 4, the addition of the regularization loss reinforces viewpoint consistency by aligning $e_{view}$ with $e_{target}$, thereby improving camera-controlled image generation. Details about regularization loss are mentioned in 4.1.5.

### 4.1.1 STEP 1: TEXT EMBEDDING OPTIMIZATION ON $I_{input}$

Initially, we enhance the CLIP embedding for $t_{input}$ with $I_{input}$ to derive $e_{optim}$ (optimized CLIP text-image embedding from $e_{input}$, which is the CLIP test embedding for $t_{input}$). This embedding

is optimized to most accurately reconstruct $I_{input}$. $f$ represents the diffusion model function that maps the input latent $x_t$, timestep $t$, and the optimized embedding $e_{optim}$. The reconstruction is achieved by minimizing the denoising diffusion loss function $L$ Ho et al. (2020), using the frozen diffusion model UNet:

$$\min_{e_{optim}} \sum_{t=T}^{0} L(f(x_t, t, e_{optim}; \theta), I_{input}) \tag{1}$$

This approach refines the text embedding to represent the characteristics of $I_{input}$ more accurately than the generic text embedding $e_{input}$.

### 4.1.2 Step 2: Fine-tuning UNet on $I_{input}$

Subsequently, the LoRA layers (with parameters $\theta_{LoRA}$) within the cross-attention layers of the diffusion model are fine-tuned at $e_{optim}$ to replicate $I_{input}$, employing the diffusion denoising loss function:

$$\min_{\theta_{LoRA}} \sum_{t=T}^{0} L(f(x_t, t, e_{optim}; \theta), I_{input}) \tag{2}$$

The rest of the UNet model remains frozen during this fine-tuning.

### 4.1.3 Step 3: Text Embedding Optimization on $I_{view}$

This process is repeated for $I_{view}$, where $e_{optim}$ is further refined to $e_{view}$ to best reconstruct $I_{view}$:

$$\min_{e_{view}} \sum_{t=T}^{0} L(f(x_t, t, e_{view}; \theta), I_{view}) \tag{3}$$

### 4.1.4 Step 4: Fine-tuning UNet on $I_{view}$ with Regularization Loss

Following the refinement of $e_{view}$, the LoRA layers are adjusted to capture the nuances of the weak guidance image $I_{view}$, guiding the transformation towards the desired viewpoint. At this stage, an additional regularization term is introduced to improve viewpoint consistency. The total loss during this step combines the reconstruction loss and the regularization loss:

$$\min_{\theta_{LoRA}} \sum_{t=T}^{0} \left( L(f(x_t, t, e_{view}; \theta), I_{view}) + L_{reg} \right) \tag{4}$$

### 4.1.5 Viewpoint Regularization

Since the CLIP model lacks an understanding of camera control information, it is essential to enhance its comprehension using 3D prior information from pretrained models. In other words, we aim to improve the viewpoint knowledge of the CLIP model by leveraging this prior knowledge, enabling it to generate the desired camera-controlled output.

We camera control results by adding a regularization term between the text embedding that includes elevation and azimuth information ($e_{target}$) and the optimized text embedding ($e_{view}$) in addition to the pretrained guidance model. In addition to enriching the viewpoint knowledge of the 3D space, applying this loss is also essential to address the limitations of the 3D-prior guidance model, as the 3D-prior model does not perform very well in complex scenes. Specifically, by applying a regularization loss between the text embedding that contains angle information and the optimized text embedding, our hypothesis is that we can improve camera control results by building a model that references the guidance image as supplementary information rather than relying solely on it.

Hence, to improve viewpoint consistency, an additional regularization loss term is added to the reconstruction loss. This term introduces a constraint between the angle embedding $e_{target}$ (representing elevation and azimuth information) and the optimized embedding $e_{view}$ during this refinement. The regularization term, calculated as $L_{reg} = \|e_{view} - e_{target}\|^2$ encourages alignment between $e_{view}$ and the intended angle information in $e_{target}$, reinforcing the target viewpoint in the generated results.

Thus, the predicted image from the 3D-based NVS method, Zero123++, serves as weak or pseudo guidance. The optimization strategy conditions the embedding space with knowledge related to the input image and its view variants using the guidance image prior which facilitates view transformation and provides a direction for viewpoint transformation.

## 4.2 INFERENCE

To generate the camera-controlled image with designated elevation ($\alpha_{elev}$) and azimuth angles ($\alpha_{azi}$), we use the target text description $t_{target}$, which varies according to the corresponding $\alpha_{elev}$ and $\alpha_{azi}$. For instance, if $\alpha_{elev} = 30°$ and $\alpha_{azi} = 30°$, $t_{target}$ can be formatted as "View from an elevated angle of +30 degrees and an azimuth angle of +30 degrees" + $t_{input}$ (e.g., "View from an elevated angle of +30 degrees and an azimuth angle of +30 degrees, An ancient Egyptian pyramid in the desert."). Next, we use the finetuned diffusion model to generate the target image using $t_{target}$, along with mutual information guidance Kothandaraman et al. (2023b), which enforces similarity between the contents of the generated and input images.

## 5 EXPERIMENTS AND RESULTS

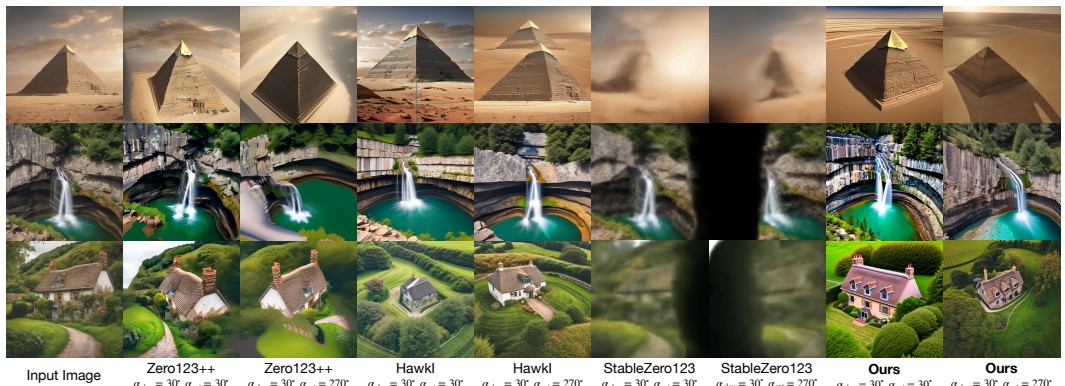

Figure 5: **Results on HawkI-Syn.** Comparisons between the state-of-the-art view synthesis models, Zero123++, HawkI, Stable Zero123, and our method highlights the superior performance of our model in terms of background inclusion, view consistency, and the accurate representation of target elevation and azimuth angles.

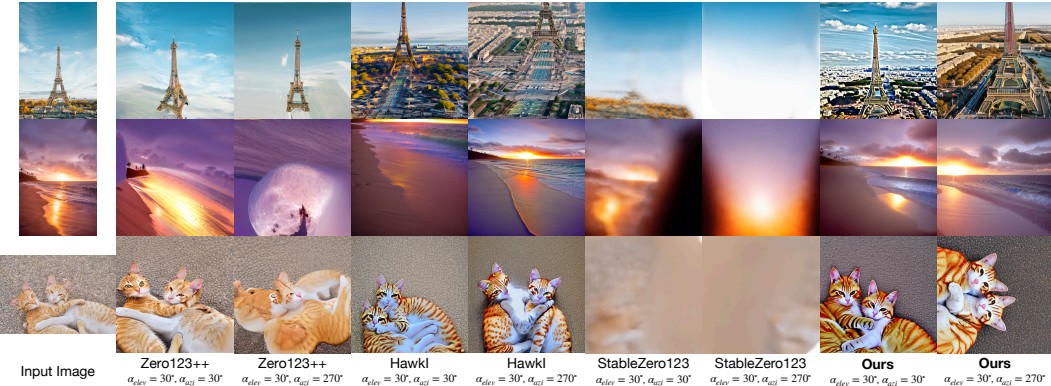

Figure 6: **Results on HawkI-Real.** Comparisons between the state-of-the-art view synthesis models, Zero123++, HawkI, Stable Zero123, and our method highlights the superior performance of our model in terms of background inclusion, view consistency, and the accurate representation of target elevation and azimuth angles.

Figure 7: **Ablation Study on the use of regularization loss between angle embedding and optimized embedding** In this experiment, we analyze the effect of adding a regularization term between the angle embedding ($e_{target}$) and the optimized embedding ($e_{view}$) on camera control results. The results show improvements in viewpoint consistency and style coherence when the regularization loss is applied.

**Datasets**    We utilize the HawkI-Syn Kothandaraman et al. (2023b) and HawkI-Real Kothandaraman et al. (2023b) datasets that feature complex scenes with multiple foreground objects and background. Both datasets provide images and text prompts to the model.

**Baselines**    We compare our method with state-of-the-art view synthesis methods: Zero123++ Shi et al. (2023a) and Stable Zero123 for 3D-based methods and HawkI Kothandaraman et al. (2023b) for 3D-free method.

**Implementation Details**    We employ the stable diffusion 2.1 model as the backbone for all our experiments and results. To generate the pseudo guidance images for different viewpoints, we use the pretrained Zero123++ Shi et al. (2023a) model. All images except those in HawkI-Real dataset are used at a resolution of $512 \times 512$. For $I_{input}$, we train the text embedding for 1,000 iterations with the learning rate of $1e - 3$ and the diffusion model for 500 iterations with the learning rate of $2e - 4$. Training the text embedding for 1,000 iterations guarantees that the text embedding $e_{optim}$ is not too close to the $e_{input}$, avoiding bias towards $I_{input}$. Likewise, it is not too distant from $e_{input}$, allowing the text embedding space to capture the characteristics of $I_{input}$. Regarding $I_{view}$, we trained the text embedding for 500 iterations and the diffusion UNet for 250 iterations. We aim for $e_{view}$ to be near $e_{optim}$ and limit the diffusion model training to 250 iterations to prevent overfitting to $I_{view}$. The purpose of $I_{view}$ is to introduce variability and provide pseudo supervision rather than accurately approximating the camera control. We set the mutual information guidance hyperparameter to $1e - 6$ and conduct inference over 50 steps.

### 5.1 QUALITATIVE ANALYSIS

We evaluate our method on four distinct viewpoints:

$$\{(\alpha_{elev}, \alpha_{azi}) \mid (30°, 30°), (-20°, 210°), (30°, 270°), (-20°, 330°)\}.$$

Our model is able to generate distinct viewpoints in the camera control to ensure consistency across generated views. Details are mentioned in the Zero123++ Shi et al. (2023a). We present qualitative representative results and comparisons with Zero123++ Shi et al. (2023a), HawkI Kothandaraman et al. (2023b), and StableZero123 at camera angles of $(30°, 30°)$ and $(30°, 270°)$ in Figures 5 and 6. Our method demonstrates superior scene reconstruction from all viewpoints compared to previous works. Specifically, results on HawkI-Syn in Figure 5 show that StableZero123 is largely ineffective. HawkI fails to capture the correct camera elevation in all cases except for the house image. While Zero123++ handles both elevation and azimuth, it struggles with background and detailed features. For instance, the pyramid in the first row lacks shadow information; the waterfall image in the second row appears unnatural; and the house in the last row blurs detailed features. Conversely, our model accurately reflects shadow characteristics in the pyramid, and reconstructs the details and background of the waterfall and house examples from various viewpoints.

Similar observations are made for HawkI-Real results shown in Figure 6. StableZero123 is ineffective. Zero123++ fails to capture background or detailed information. For example, when tasked

with camera control for an image of the Eiffel Tower, Zero123++ focuses solely on the Eiffel Tower, ignoring surrounding details. The original HawkI model, while producing aerial views, fails in angle conversion tasks. In contrast, our model accurately performs angle conversion tasks at $(30°, 30°)$ and $(30°, 270°)$, including the Seine River in the background for the Eiffel Tower image, showcasing its superiority. Camera control tasks for the HawkI-Real dataset, including images like the Hawaii beach and a cat, further demonstrate our model's excellence compared to other models. The key benefits of our model over 3D-based NVS methods such as Zero123++ and 3D-free methods such as HawkI arises by merging the strengths of 3D-based techniques into a 3D-free optimization process, effectively combining the best features of both.

## 5.2 QUANTITATIVE EVALUATION

Following prior work Shi et al. (2023a); Kothandaraman et al. (2023b); Liu et al. (2023a), we evaluate our method using six metrics - (i) LPIPS Zhang et al. (2018): Quantifies the perceptual similarity between the generated and input images, with lower values indicating better performance. (ii) CLIP-Score Radford et al. (2021) and View-CLIP Score: Measure text-based alignment of the generated images. The CLIP score assesses alignment with both content and the $(\alpha_{elev}, \alpha_{azi})$ viewpoint, while the View-CLIP Score focuses specifically on the viewpoint. Higher values are preferred. (iii) DINO Caron et al. (2021), SSCD Pizzi et al. (2022): DINO evaluates the semantic consistency of the generated images by comparing high-level feature embeddings extracted from a self-supervised vision transformer. DINO is trained not to ignore differences between subjects of the same class. Higher values indicate better preservation of semantic content across different views of the same scene. SSCD measures structural similarity between the generated images and their reference counterparts using learned feature representations. SSCD focuses on capturing fine-grained structural and contextual consistency. Higher values are preferred for better alignment with ground-truth structures. (iv) CLIP-I Ruiz et al. (2023): CLIP-I measures the cosine similarity between the embeddings of multi-view images and the input image within the CLIP space. (v) PSNR (Peak Signal-to-Noise Ratio) and SSIM Wang et al. (2004) (Structural Similarity Index): PSNR Quantifies the pixel-wise fidelity of the generated images relative to the reference images. PSNR is calculated as the logarithmic ratio of the maximum possible pixel value to the mean squared error between the two images. Higher values indicate better pixel-level accuracy. SSIM assesses perceptual similarity by comparing luminance, contrast, and structural information between the generated and reference images. SSIM is designed to measure structural consistency, with higher values reflecting closer perceptual alignment.

Similar to the quantitative comparison performed by Zero123++, we use 10% of the overall data from the HawkI-Syn and HawkI-Real datasets as the validation set to compute the quantitative metrics. Table 1 and Table 5 shows that our model significantly outperforms the state-of-the-art across these evaluation metrics, reinforcing how our models stands out by incorporating the robust features of 3D-based NVS methods into a 3D-free optimization strategy, thereby capitalizing on the benefits of both approaches.

| Dataset, Angle, Method | LPIPS ↓ | CLIP ↑ | DINO ↑ | SSCD ↑ | CLIP-I ↑ | PSNR ↑ | SSIM ↑ |
|---|---|---|---|---|---|---|---|
| HawkI-Syn $(30°, 30°)$ Ours | **0.5661** | **29.9563** | **0.4314** | 0.3638 | **0.8317** | **11.0664** | **0.3162** |
| HawkI-Syn $(30°, 30°)$ HawkI | 0.5998 | 28.3786 | 0.3982 | 0.3519 | 0.8221 | 10.7092 | 0.2941 |
| HawkI-Syn $(30°, 30°)$ Zero123++ | 0.5694 | 28.2555 | 0.4293 | **0.4605** | 0.8149 | 10.9923 | 0.3073 |
| HawkI-Syn $(30°, 30°)$ Stable Zero123 | 0.7178 | 21.3430 | 0.2108 | 0.2386 | 0.6467 | 9.2585 | 0.1954 |
| HawkI-Syn $(30°, 270°)$ Ours | **0.5744** | **29.1800** | **0.4148** | **0.3684** | **0.8327** | **11.0661** | **0.3047** |
| HawkI-Syn $(30°, 270°)$ HawkI | 0.5971 | 27.9540 | 0.3964 | 0.3473 | 0.8278 | 10.6303 | 0.2779 |
| HawkI-Syn $(30°, 270°)$ Zero123++ | 0.6056 | 25.6665 | 0.2681 | 0.2195 | 0.7087 | 10.4395 | 0.2984 |
| HawkI-Syn $(30°, 270°)$ Stable Zero123 | 0.6785 | 23.1555 | 0.2119 | 0.2657 | 0.6456 | 9.4703 | 0.1673 |
| HawkI-Real $(30°, 30°)$ Ours | **0.6201** | **29.8850** | **0.3346** | 0.2588 | **0.8152** | **9.4009** | **0.2184** |
| HawkI-Real $(30°, 30°)$ HawkI | 0.6529 | 27.5847 | 0.2844 | 0.2269 | 0.7754 | 8.9257 | 0.2160 |
| HawkI-Real $(30°, 30°)$ Zero123++ | 0.6253 | 27.9877 | 0.3315 | **0.3362** | 0.8023 | 9.2962 | 0.1990 |
| HawkI-Real $(30°, 30°)$ Stable Zero123 | 0.6614 | 23.0895 | 0.1781 | 0.1192 | 0.6569 | 7.7977 | 0.1684 |
| HawkI-Real $(30°, 270°)$ Ours | **0.5868** | **30.5489** | **0.4126** | **0.3424** | **0.8708** | 10.6177 | **0.2687** |
| HawkI-Real $(30°, 270°)$ HawkI | 0.6215 | 29.0488 | 0.3530 | 0.3363 | 0.8358 | **10.6472** | 0.2439 |
| HawkI-Real $(30°, 270°)$ Zero123++ | 0.6302 | 27.5228 | 0.3145 | 0.2005 | 0.7529 | 9.8864 | 0.2484 |
| HawkI-Real $(30°, 270°)$ Stable Zero123 | 0.6268 | 21.1090 | 0.1750 | 0.0494 | 0.6500 | 8.3163 | 0.1637 |

Table 1: **Quantitative Results**. Evaluation of seven metrics demonstrates the superior results of our method over prior work.

| Dataset, Angle, Method | LPIPS ↓ | CLIP ↑ | DINO ↑ | SSCD ↑ | CLIP-I ↑ | PSNR ↑ | SSIM ↑ |
|---|---|---|---|---|---|---|---|
| HawkI-Syn $(30°, 30°)$ w/ regularization | **0.5661** | **29.9563** | **0.4314** | 0.3638 | **0.8317** | **11.0664** | **0.3162** |
| HawkI-Syn $(30°, 30°)$ w/o regularization | 0.5867 | 28.5417 | 0.4122 | **0.3640** | 0.8243 | 10.8272 | 0.2954 |
| HawkI-Real $(30°, 30°)$ w/ regularization | **0.6201** | **29.8850** | 0.3346 | **0.2588** | 0.8152 | **9.4009** | **0.2184** |
| HawkI-Real $(30°, 30°)$ w/o regularization | 0.6257 | 29.0798 | **0.3357** | 0.2401 | **0.8231** | 9.1957 | 0.2014 |
| HawkI-Syn $(30°, 270°)$ w/ regularization | **0.5744** | **29.1800** | **0.4148** | **0.3684** | **0.8327** | **11.0661** | **0.3047** |
| HawkI-Syn $(30°, 270°)$ w/o regularization | 0.5952 | 28.9866 | 0.4098 | 0.3350 | 0.8248 | 10.8656 | 0.2850 |
| HawkI-Real $(30°, 270°)$ w/ regularization | **0.5868** | **30.5489** | **0.4126** | **0.3424** | **0.8708** | **10.6177** | **0.2687** |
| HawkI-Real $(30°, 270°)$ w/o regularization | 0.6114 | 29.9184 | 0.4003 | 0.3075 | 0.8541 | 10.2958 | 0.2615 |
| HawkI-Syn $(-20°, 210°)$ w/ regularization | **0.5740** | **29.1144** | **0.4277** | 0.3529 | **0.8280** | **10.9697** | **0.2837** |
| HawkI-Syn $(-20°, 210°)$ w/o regularization | 0.5860 | 28.6385 | 0.3969 | **0.3559** | 0.8171 | 10.8401 | 0.2792 |
| HawkI-Real $(-20°, 210°)$ w/ regularization | **0.6185** | **30.6729** | 0.3610 | **0.2880** | **0.8448** | **10.3130** | **0.2223** |
| HawkI-Real $(-20°, 210°)$ w/o regularization | 0.6338 | 29.1693 | **0.3817** | 0.2605 | 0.8263 | 10.0794 | 0.2117 |
| HawkI-Syn $(-20°, 330°)$ w/ regularization | **0.5624** | **29.2144** | **0.4487** | **0.3892** | **0.8559** | **11.2175** | **0.3048** |
| HawkI-Syn $(-20°, 330°)$ w/o regularization | 0.5714 | 28.4089 | 0.4476 | 0.3870 | 0.8492 | 11.0409 | 0.2947 |
| HawkI-Real $(-20°, 330°)$ w/ regularization | 0.5925 | **29.5090** | **0.3899** | **0.3127** | **0.8689** | **10.6183** | **0.2971** |
| HawkI-Real $(-20°, 330°)$ w/o regularization | **0.5894** | 28.8531 | 0.3704 | 0.2954 | 0.8506 | 10.5213 | 0.2828 |

Table 2: **Quantitative Results of Ablation Study**. Evaluation of seven metrics demonstrates the superior results of the regularized method over the non-regularized one.

## 5.3 ABLATION ANALYSIS: VIEWPOINT REGULARIZATION LOSS

To demonstrate the effectiveness of our approach in achieving camera control, we present results for scenes such as a Hawaiian beach and a waterfall. In both instances, the guidance images from Zero123++ fail to provide accurate direction to the model. For the Hawaiian beach scene, the output generated with the regularization term exhibits a more consistent style compared to the output produced without it. Despite the inaccuracies in the Zero123++ guidance images, the regularization term facilitates more reliable camera control than the results generated without it.

Similarly, in the waterfall scene, the regularization term enhances the consistency of the generated rock textures surrounding the waterfall. Without the regularization term, these textures are inconsistently represented; however, with it, the style is maintained more faithfully. Once again, Zero123++ does not provide accurate guidance in this case, underscoring the significant contribution of the regularization loss to improved control and visual coherence in the generated images. Detailed results from our ablation study are presented in Figure 7. Furthermore, the application of the regularization loss demonstrates performance improvements in quantitative evaluations, as shown in Table 2.

## 6 CONCLUSIONS, LIMITATIONS AND FUTURE WORK

In this paper, we propose an approach that integrates the advantages of off-the-shelf 3D-based pretrained models within 3D-free paradigms for novel view synthesis, offering precise control over camera angle and elevation, without any additional 3D information. Our method performs effectively on complex, in-the-wild images containing multiple objects and background information. We qualitatively and quantitatively demonstrate the benefits of our method over corresponding 3D and 3D-free baselines. One limitation of our method is its reliance on an inference-time optimization routine for each scene and viewpoint, which may hinder real-time performance. Achieving faster performance is a direction for future work. Additionally, extending our approach to NVS and 3D applications with real-world constraints (such as respecting contact points and relative sizes) for tasks like editing, object insertion, and composition presents promising directions for further research. Based on the current results, we also propose exploring the use of an image-conditioned model to achieve a higher level of view consistency as a future research direction. As mentioned in the qualitative analysis, Due to its design, Zero123++ is limited to generating only distinct fixed views. While this approach improves consistency by leveraging Stable Diffusion's priors, it restricts the model's ability to generate views beyond these predefined angles, limiting flexibility in exploring arbitrary perspectives. Future work could explore enabling camera control from any angle, while addressing 3D prior model's challenges like preserving source view attributes and mitigating issues from incorrect pose information to improve consistency and accuracy.

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

# A  APPENDIX

## A.1  ADDITIONAL RESULTS

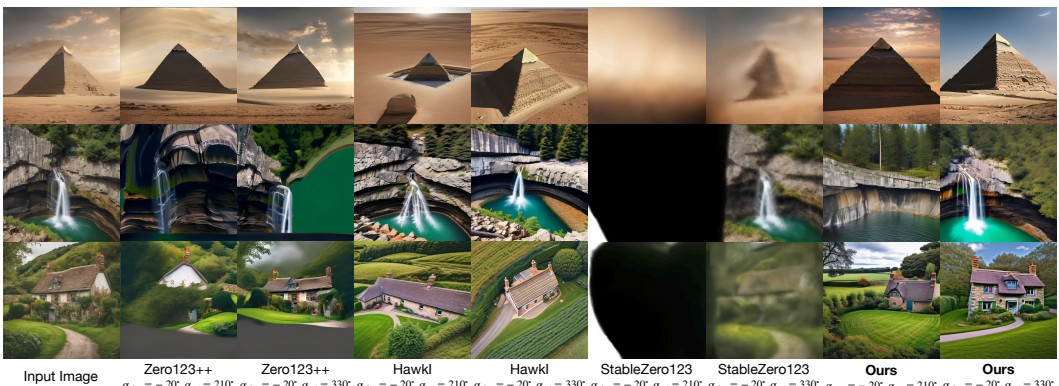

Figure 8: **More results on HawkI-Syn.** We present additional comparison results on HawkI-Syn for the angles of $(-20°, 210°)$ and $(-20°, 330°)$. Our model consistently produces view synthesis images that maintained background inclusion and view consistency, accurately mirroring the target elevation and azimuth angles. Notably, StableZero123 exhibits instability in its results. It's important to highlight that this task specifically addresses *negative azimuth angles*. HawkI, for instance, fails to capture the correct camera elevation and is limited to generating aerial views. Zero123++ is capable of handling both elevation and azimuth but falls short in integrating background elements and intricate details, as also observed in previous outcomes. For example, when presented with an image of a pyramid casting a shadow, Zero123++ darkens the pyramid but fails to render the shadow accurately. This shortcoming is also apparent in images of a waterfall and a house. In the waterfall task within the specified azimuth range, Zero123++ produces an indistinct shape rather than a clear environment where water and lake are visible from below the rocks. Similarly, for the house image, it generates an incomplete image with gray patches. Conversely, our model not only captures the shadow details of the pyramid but also accurately renders the environment in the waterfall image, ensuring visibility of water and lake from beneath the rocks. Additionally, it adeptly incorporates details and backgrounds from multiple perspectives.

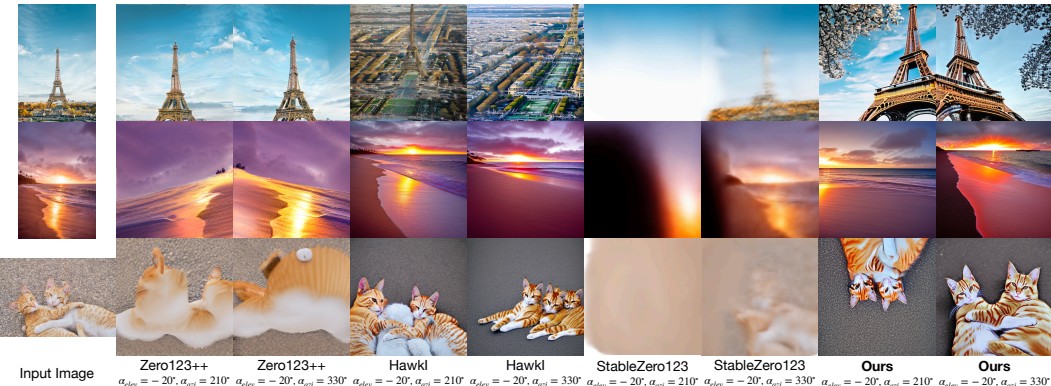

Figure 9: **More Results on HawkI-Real.** We extend our analysis to additional settings of $(-20°, 210°)$ and $(-20°, 330°)$. Our model, when tested on the HawkI-Real dataset, demonstrated superior performance in view synthesis images, excelling in background inclusion and view consistency, and accurately representing the target elevation and azimuth angles. In comparison to other leading models such as Zero123++, HawkI, and StableZero123, our model's results are notably better. StableZero123's outputs are incomplete, and Zero123++ struggles with capturing background details and intricate information. Specifically, Zero123++ neglected surrounding details, focusing solely on the Eiffel Tower. The original HawkI model also failed to achieve the correct camera elevation or produced images that overlooked important features. For example, in the cat transformation task, the output incorrectly depicted three cats instead of two. Our model stands out by delivering exceptional results for the Eiffel Tower, Hawaiian beach, and cat transformations, underscoring its advanced capabilities over other models. Furthermore, we present a quantitative evaluation in Table 5, which confirms our model's dominance over state-of-the-art benchmarks across various metrics.

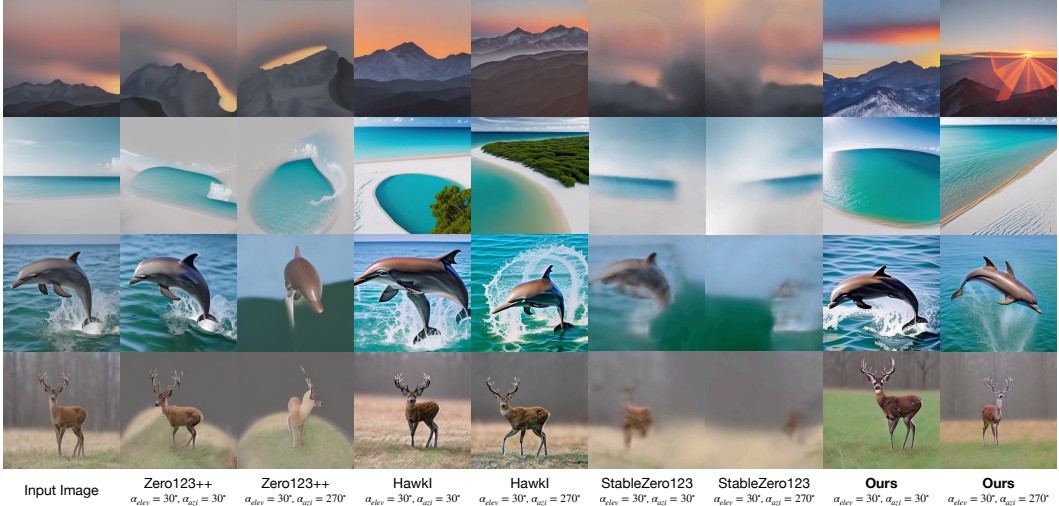

Figure 10: Additional comparisons in $(30°, 30°)$ and $(30°, 270°)$ settings on images from the HawkI-Syn and HawkI-Real datasets. Comparisons between the state-of-the-art view synthesis models, Zero123++, HawkI, Stable Zero123, and our method highlights the superior performance of our model in terms of background inclusion, view consistency, and the accurate representation of target elevation and azimuth angles.

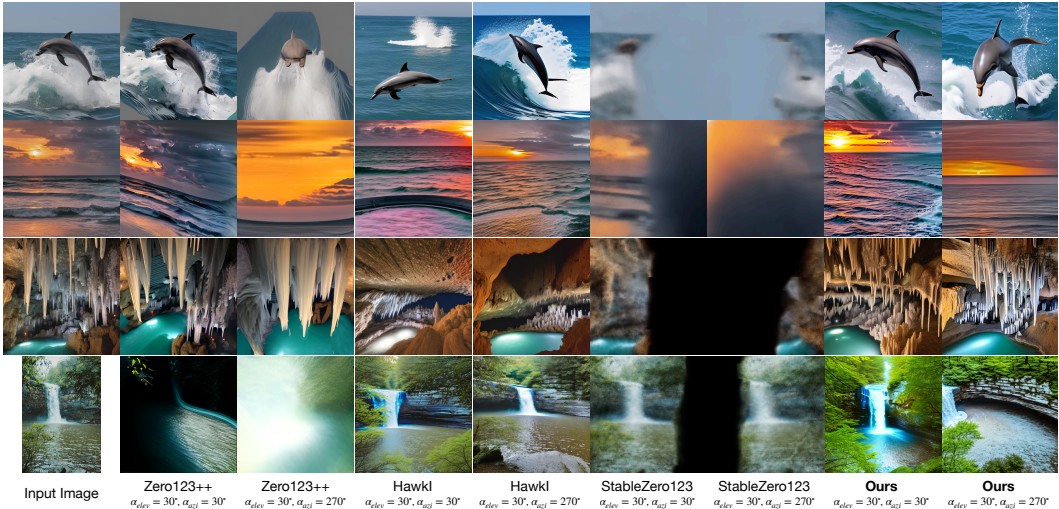

Figure 11: Additional comparisons in $(30°, 30°)$ and $(30°, 270°)$ settings on images from the HawkI-Syn and HawkI-Real datasets. Comparisons between the state-of-the-art view synthesis models, Zero123++, HawkI, Stable Zero123, and our method highlights the superior performance of our model in terms of background inclusion, view consistency, and the accurate representation of target elevation and azimuth angles.

Figure 12: Additional comparisons in $(30°, 30°)$ and $(30°, 270°)$ settings on images from the HawkI-Syn and HawkI-Real datasets. Comparisons between the state-of-the-art view synthesis models, Zero123++, HawkI, Stable Zero123, and our method highlights the superior performance of our model in terms of background inclusion, view consistency, and the accurate representation of target elevation and azimuth angles.

## A.2 COMPUTATION TIME

Table 3 presents a comparison of memory consumption and computation time across state-of-the-art 3D-prior models, including Zero123++, Stable Zero123, and ZeroNVS. Among these models, Zero123++ demonstrates the shortest computation time, requiring only 20 seconds, while other methods are significantly slower.

Our approach utilizes Zero123++ for generating 3D prior information, ensuring that the computational cost remains minimal. Importantly, the generation of multi-view guidance images does not introduce any additional overhead, as this step is performed using the most computationally efficient

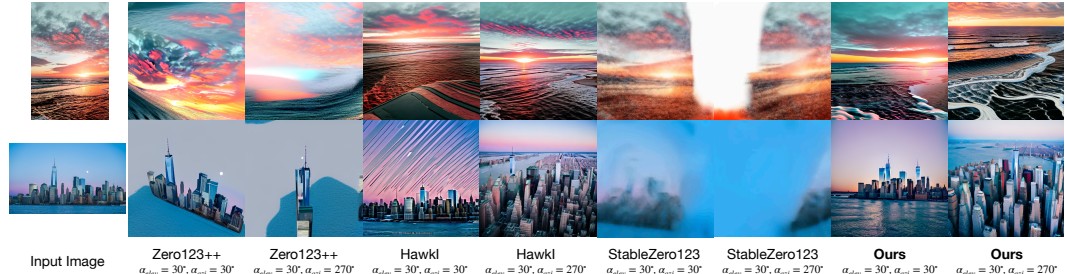

Figure 13: Additional comparisons in $(30°, 30°)$ and $(30°, 270°)$ settings on images from the HawkI-Syn and HawkI-Real datasets. Comparisons between the state-of-the-art view synthesis models, Zero123++, HawkI, Stable Zero123, and our method highlights the superior performance of our model in terms of background inclusion, view consistency, and the accurate representation of target elevation and azimuth angles.

model in this category. This demonstrates that our method is well-suited for scalable and real-time applications, maintaining efficiency while incorporating powerful 3D prior information.

| Model | Memory Consumption | Computation Time |
|---|---|---|
| Zero123++ | 10.18 GB / 40.0 GB (9,715 MiB) | 20 sec |
| Stable Zero123 | 39.3 GB / 40.0 GB (37,479 MiB) | 1,278 sec |
| ZeroNVS | 33.48 GB / 40.0 GB (31,929 MiB) | 7,500 sec |

Table 3: **Comparison of computation times for 3D-prior models.** Among the prior works in NVS frequently mentioned, including Zero123++, Stable Zero123, and ZeroNVS, the Zero123++ model has the shortest computation time. Our research applies the Zero123++ model, which has the lowest computation time among 3D-prior models, to obtain 3D prior information without requiring any additional computation time.

Table 4 provides a detailed breakdown of memory usage and computation time for the optimization and inference steps in our method. The optimization process requires 387 seconds (Optimization 367 sec + Zero123++ 20 sec), and the inference step is highly efficient, taking only 6 seconds per image. Notably, the memory consumption remains consistent across optimization and inference, excluding the Zero123++ computation, comparable to other competitive methods.

This breakdown highlights that the inclusion of the Zero123++ step in our approach does not result in excessive computational time. Instead, our method achieves high-quality multi-view synthesis while maintaining practical memory and runtime efficiency. Furthermore, the results illustrate that our approach is capable of integrating multi-view guidance and reconstructing images with enhanced fidelity without compromising scalability or practicality.

This experiment was conducted using an A100 GPU (40.0 GB) for all models to ensure a fair comparison. The GPU memory consumption for each model is reported in the worst-case scenario, and the computation time is measured based on the time taken for the model to fully generate an image. The results in Table 3 and Table 4 emphasize that the proposed method effectively balances computational efficiency with enhanced performance. By leveraging Zero123++, the most efficient 3D-prior model, and incorporating lightweight optimization techniques, our approach ensures minimal computational costs while achieving significant improvements in output quality. These results validate the feasibility of the method for real-world applications, demonstrating both its scalability and practicality.

| Model | Step | Memory Consumption | Computation Time |
|---|---|---|---|
| HawkI | Optimization | 7.20 GB / 40.0 GB (6,875 MiB) | 395 sec |
| | Inference | 8.43 GB / 40.0 GB (8,045 MiB) | 6 sec (each image) |
| Total | | **8.43 GB** / 40.0 GB (8,045 MiB) | **401 sec** |
| Ours (w/o regloss) | Zero123++ | 10.18 GB / 40.0 GB (9,715 MiB) | 20 sec |
| | Optimization | 7.21 GB / 40.0 GB (6,879 MiB) | 372 sec |
| | Inference | 8.43 GB / 40.0 GB (8,049 MiB) | 6 sec (each image) |
| Total | | **10.18 GB** / 40.0 GB (9,715 MiB) | **398 sec** |
| Ours (w/ regloss) | Zero123++ | 10.18 GB / 40.0 GB (9,715 MiB) | 20 sec |
| | Optimization | 7.21 GB / 40.0 GB (6,885 MiB) | 367 sec |
| | Inference | 8.43 GB / 40.0 GB (8,045 MiB) | 6 sec (each image) |
| Total | | **10.18 GB** / 40.0 GB (9,715 MiB) | **393 sec** |

Table 4: **Detailed Step-wise Comparison**. Even when applying Zero123++ to our methodology, the additional GPU memory consumption is relatively small, at **2.97GB** (10.18GB - 7.21GB), and it takes only **20 seconds** to generate the guidance image using Zero123++. From an overall perspective, HawkI takes 401 seconds to complete optimization and generate the first image through inference, while Ours (w/o regloss) takes 398 seconds, and Ours (w/ regloss) takes **393 seconds**. This demonstrates that our methodology does not result in significant differences in computation time or memory consumption while achieving better performance compared to existing methods. Total memory consumption refers to the worst case, the computation time indicates the total execution time. *i.e.,* the time taken for the model to run and output the first image.

## A.3 QUANTITATIVE EVALUATION RESULTS

| Dataset, Angle, Method | LPIPS ↓ | CLIP ↑ | DINO ↑ | SSCD ↑ | CLIP-I ↑ | PSNR ↑ | SSIM ↑ |
|---|---|---|---|---|---|---|---|
| HawkI-Syn $(-20°, 210°)$ Ours | **0.5740** | **29.1144** | **0.4277** | **0.3529** | **0.8280** | **10.9697** | **0.2837** |
| HawkI-Syn $(-20°, 210°)$ HawkI | 0.6024 | 27.7407 | 0.3831 | 0.3494 | 0.8226 | 10.5667 | 0.2744 |
| HawkI-Syn $(-20°, 210°)$ Zero123++ | 0.6037 | 24.4148 | 0.2936 | 0.3021 | 0.7309 | 10.7458 | 0.2803 |
| HawkI-Syn $(-20°, 210°)$ Stable Zero123 | 0.7452 | 20.7860 | 0.0852 | 0.0996 | 0.5634 | 6.3887 | 0.0971 |
| HawkI-Syn $(20°, 330°)$ Ours | **0.5624** | **29.2144** | **0.4487** | 0.3892 | **0.8559** | **11.2175** | **0.3048** |
| HawkI-Syn $(20°, 330°)$ HawkI | 0.5943 | 27.5738 | 0.4080 | 0.3532 | 0.8152 | 10.8882 | 0.2759 |
| HawkI-Syn $(20°, 330°)$ Zero123++ | 0.5652 | 25.8831 | 0.4305 | **0.4431** | 0.7932 | 11.1130 | 0.2936 |
| HawkI-Syn $(20°, 330°)$ Stable Zero123 | 0.6332 | 23.2087 | 0.3366 | 0.3393 | 0.6890 | 9.1852 | 0.1943 |
| HawkI-Real $(-20°, 210°)$ Ours | **0.6185** | **30.6729** | **0.3610** | **0.2880** | **0.8448** | **10.3130** | **0.2223** |
| HawkI-Real $(-20°, 210°)$ HawkI | 0.6464 | 28.7500 | 0.3567 | 0.2697 | 0.8001 | 9.6859 | 0.2145 |
| HawkI-Real $(-20°, 210°)$ Zero123++ | 0.6816 | 24.7083 | 0.2101 | 0.1706 | 0.6434 | 8.6865 | 0.2194 |
| HawkI-Real $(-20°, 210°)$ Stable Zero123 | 0.6650 | 21.5791 | 0.1564 | 0.0225 | 0.5850 | 7.4097 | 0.1681 |
| HawkI-Real $(-20°, 330°)$ Ours | **0.5925** | **29.5090** | **0.3899** | **0.3127** | **0.8689** | **10.6183** | **0.2971** |
| HawkI-Real $(-20°, 330°)$ HawkI | 0.6283 | 27.5200 | 0.3228 | 0.2406 | 0.8383 | 10.4706 | 0.2787 |
| HawkI-Real $(-20°, 330°)$ Zero123++ | 0.5978 | 26.1550 | 0.3735 | 0.3080 | 0.8043 | 10.5917 | 0.2953 |
| HawkI-Real $(-20°, 330°)$ Stable Zero123 | 0.6673 | 25.6611 | 0.2667 | 0.1998 | 0.7249 | 9.0786 | 0.1653 |

Table 5: **Quantitative Results**. Evaluation of seven metrics demonstrates the superior results of our method over prior work.

