# OpenReview forum: "3D-free meets 3D priors: Novel View Synthesis from a Single Image with Pretrained Diffusion Guidance"
_ICLR.cc/2025/Conference — Submitted to ICLR 2025_

### Official Review · Reviewer_yRXV · 2024-10-19

**Soundness:** 2
**Presentation:** 1
**Contribution:** 1
**Rating:** 3
**Confidence:** 4

**Summary:**

This paper argues that current NVS methods are limited to single-object-centric scenes and struggle with complex environments. Moreover, they require extensive 3D data for training. Therefore, this paper proposes a training-free method to combine the benefits of 3D-free and 3D-based methods. It leverages pretrained NVS models for weak guidance and integrates the knowledge into a 3D-free approach to achieve the desired results.

**Strengths:**

1. The motivation for training-free novel view synthesis is good.
2. The analysis of the current NVS model’s limitation is reasonable.

**Weaknesses:**

1. The writing is poor and many sentences are extremely confusing without any support. I will list several examples below and there are more similar problems in the submitted paper.

    In Line 265. “Explain the subset you used for your results, how you created the,….”. What does the author want to express here?

    In section 3.1, why does Fig.3 illustrate these findings? I believe the findings here are that CLIP struggles with 3D comprehension, but I do not see any examples in Fig.3 supporting this viewpoint. Moreover, if CLIP itself can control camera pose change, there is no need for training on 3D-aware data pairs. I do not see the necessity of including this discussion.

    In Line 146. “validates the necessity of 3D guidance images.” What is 3D guidance images? Is it a novel view image based on the current image?

    In Line 159. “This indicates that the model benefits from the information in the 3D-derived guidance image, and that the guidance image is not simply a source of extra variance.” What does the author want to express here? What is the 3D-derived guidance image? Does it also mean a novel view image? If so, what does the author mean by “not simply a source of extra variance”? What is the variance? From Fig.4, I guess the author is conducting an experiment on using different novel view images as weak guidance while prompting the generation of novel view images using the proposed training-free method. The reviewer thinks using the right guidance image is quite trivial.

2. The performance improvement is weak. From the visualized comparison in Fig.5, I don’t see much difference between Zero123++ and Ours. Is the camera view control more precise or is the image more realistic? From the waterfall example, I think the result of Zero123++ is more realistic.
3. The novelty is weak and the method is not well explained. I think a training-free method is a good motivation. But the author still requires heavily on a pretrained NVS model.
4. Related work is not well discussed. To cope with the complexity of real scene instead of object-level, many works have been proposed and the author should include them. I will list several below.

[1] ZeroNVS: Zero-Shot 360-Degree View Synthesis from a Single Real Image, CVPR 2024

[2] MegaScenes: Scene-Level View Synthesis at Scale, ECCV 2024

[3] Generative Camera Dolly: Extreme Monocular Dynamic Novel View Synthesis, ECCV 2024

**Questions:**

Minor:
In Tab.1 and Tab.2, since the author reports the LPIPS metric, I believe the commonly used PSNR and SSIM metrics can also be reported.

The reviewer suggests that the author revise the writing, conduct more thorough experiments, and propose a clearer explanation of motivation and method.

---

> ### Author Response · Authors · 2024-11-22
> **Rebuttal by Authors**
>
> Thank you sincerely for your constructive and detailed feedback. We fully understand the points you have raised and have provided responses below.
>
> > **(1) The writing is poor and many sentences are extremely confusing without any support. I will list several examples below and there are more similar problems in the submitted paper. / In Line 265. (...) / In Line 146. (...) / In Line 159. (...)**
>
> Thank you for pointing that out. We have identified the typo you mentioned and corrected it accordingly.
>
> > **(2) In section 3.1, why does Fig.3 illustrate these findings? I believe the findings here are that CLIP struggles with 3D comprehension, but I do not see any examples in Fig.3 supporting this viewpoint. Moreover, if CLIP itself can control camera pose change, there is no need for training on 3D-aware data pairs. I do not see the necessity of including this discussion.**
>
> Section 3.1 describes the process of generating images solely from a text prompt, excluding the guidance image required by the base HawkI model while providing angle-related information. As you can see, the resulting images are inconsistent, and the camera control angles are not properly followed.
>
> One example is elevation. Specifically, as shown in Figure 3, when the elevation was set to -20, the model still generated a view from an aerial perspective and failed to reflect the elevation value of -20. This demonstrates that the CLIP included in the View Synthesis method, HawkI, is still not perfect.
>
> Hence, our hypothesis is that the **guidance image plays a critical role.** This naturally leads to the question of why the guidance image is important.
>
> This experiment highlights the importance of guidance images and plays a significant role in shaping the main idea of this paper, which involves providing guidance images using a 3D-prior model and applying regularization loss.
>
> > **(2) The performance improvement is weak. From the visualized comparison in Fig.5, I don’t see much difference between Zero123++ and Ours. Is the camera view control more precise or is the image more realistic? From the waterfall example, I think the result of Zero123++ is more realistic.**
>
> Please refer to A2 of the global rebuttal at the top!
>
> > **(3) The novelty is weak and the method is not well explained. I think a training-free method is a good motivation. But the author still requires heavily on a pretrained NVS model.**
>
> Please refer to A2 of the global rebuttal at the top! There have been improvements in the results/methodology.
>
> > **(4) Related work is not well discussed. To cope with the complexity of real scene instead of object-level, many works have been proposed and the author should include them. I will list several below. / [1] ZeroNVS: Zero-Shot 360-Degree View Synthesis from a Single Real Image, CVPR 2024 / [2] MegaScenes: Scene-Level View Synthesis at Scale, ECCV 2024 / [3] Generative Camera Dolly: Extreme Monocular Dynamic Novel View Synthesis, ECCV 2024**
>
> Thank you for your insightful feedback. We have added the three papers you mentioned to the Related Works and Introduction sections, along with additional related works.
>
> > **(5) Minor: In Tab.1 and Tab.2, since the author reports the LPIPS metric, I believe the commonly used PSNR and SSIM metrics can also be reported.**
>
> Thank you for the valuable feedback. We have **revised the evaluation metrics and included PSNR and SSIM as part of the updates.** We followed the evaluation approach of the baseline model, HawkI. However, metrics like View-CLIP, CLIPD, and View-CLIPD, while relevant to our task, were deemed somewhat mismatched (and potentially unfair), so we decided to exclude them. Instead, we added metrics commonly used in other view synthesis tasks, such as **PSNR and SSIM, as well as DINO and SSCD**, to enhance the reliability of our evaluation. The **LPIPS metric used by Zero123++** has already been included in the previous paper we submitted.
>
> > **(6) The reviewer suggests that the author revise the writing, conduct more thorough experiments, and propose a clearer explanation of motivation and method.**
>
> Please refer to A2 of the global rebuttal at the top!
>
> There have been **significant improvements in the results and the paper, and based on the positive evaluations from several reviewers regarding the novelty and potential of our idea,** we would like to kindly and respectfully ask if you could consider raising the review rating for our paper.
>
> Thank you sincerely for taking the time to provide such detailed reviews. If you have any additional questions, please feel free to reach out at any time!

---

> > ### Comment · Area_Chair_5oV2 · 2024-11-24
> > **Discussion Period Ending Soon**
> >
> > Dear Reviewer,
> >
> > The discussion period will end soon. Please take a look at the author's comments and begin a discussion.
> >
> > Thanks, Your AC

---

> > ### Comment · Reviewer_yRXV · 2024-11-26
> >
> > I appreciate the authors' efforts during the rebuttal, particularly their revisions addressing some of the errors pointed out by reviewers. However, several of my concerns remain unresolved:
> >
> > 1. **Consistency with the input view**: Consistency with the input view is crucial for novel view synthesis. This should have been addressed by incorporating the 3D prior (Zero123++) and fine-tuning Stable Diffusion, as proposed in the paper. However, based on the results presented (e.g., Fig. 5 and Fig. 6), the generated images show poor consistency with the input view and fail to maintain the original structure.
> > 2. **Oversaturation issues**: The generated images suffer from oversaturation (e.g., Fig. 1, Fig. 5, and Fig. 6), with colors being overly vivid and intense, deviating significantly from the original images. I believe such results difficult to accept for novel view synthesis tasks.
> > 3. **Limited improvements over baselines**: Despite the additional experiments provided during the rebuttal, the improvements over baselines remain minimal. Notably, the best PSNR reported in the paper does not exceed 11.5 dB, with a maximum improvement over baselines of less than 0.5 dB. Such marginal gains are quite weak, and I suggest that the authors further optimize their approach to achieve more competitive results before resubmitting.
> > 4. **Application limitations**: As far as I know, Zero123++ is restricted to generating fixed viewpoints. Building on this prior significantly limits the applicability of the proposed method. I recommend the authors address this limitation.
> > 5. **Writing issues**: Although some corrections were made, there are still noticeable errors in the paper. For instance, inconsistent notation is used (e.g., in Eq. 3 of Sec. 4.1.3, $e_{\text{view}}$ is presented as plain text, whereas $I_{view}$ is in mathematical notation). Additionally, in line L137, a reference contains a question mark. These issues indicate that the writing quality does not meet the standard for acceptance. I hope the authors will address these problems in future versions.
> >
> > In conclusion, I still believe this paper is not ready for acceptance. Therefore, I will maintain my original reject rating.

---

> > > ### Author Response · Authors · 2024-11-27
> > > **Official Comment by Authors (1)**
> > >
> > > Dear Reviewer yRXV,
> > >
> > > We sincerely thank you for your thoughtful feedback and for highlighting important points.
> > >
> > > > **1. Consistency with the input view & 3. Limited improvements over baselines**
> > >
> > > Thank you for your valuable feedback. This point aligns with the feedback we received from BRTu reviewers. As you mentioned, we addressed the issue by integrating the 3D-prior model, Zero123++. To resolve the consistency-related concerns raised in the initial reviews, **we utilized a regularization loss and conducted both quantitative and qualitative evaluations.** The details of this are presented in Section 5.3, Figure 7, and Table 2.
> > >
> > > In particular, the examples highlighted in the ablation study shown in Figure 7 include scenes of a Hawaiian beach and a waterfall. In both cases, the Zero123++ model failed to provide accurate viewpoints and guidance to the model. However, the application of the regularization loss **significantly improved the consistency of the textures in the Hawaiian beach scene and enhanced the consistency of the rock textures generated in the waterfall scene.**
> > >
> > > We kindly ask you to refer to Section 5.3 for more detailed information on this aspect.
> > >
> > > > *Part of sentences from Section 5.3*: "For the Hawaiian beach scene, the **output generated with the regularization term exhibits a more consistent style** compared to the output produced without it. Despite the inaccuracies in the Zero123++ guidance images, the regularization term facilitates **more reliable camera control than the results generated without it.**
> > > Similarly, in the waterfall scene, the regularization term enhances the consistency of the generated rock textures surrounding the waterfall. **Without the regularization term, these textures are inconsistently represented; however, with it, the style is maintained more faithfully.** Once again, Zero123++ does not provide accurate guidance in this case, underscoring the significant contribution of the regularization loss to improved control and visual coherence in the generated images."
> > >
> > > We also conducted an **ablation study** to compare the results with and without the application of the regularization loss. By analyzing seven image evaluation metrics presented in the existing evaluation — LPIPS, CLIP, DINO, SSCD, CLIP-I, PSNR, and SSIM — we demonstrated superior performance, thereby providing quantitative evaluations. We have attached the results in a table below for your reference, and we would appreciate it if you could take a look.

---

> > > > ### Author Response · Authors · 2024-11-27
> > > > **Official Comment by Authors (2)**
> > > >
> > > > > | Dataset, Angle, Method                       | LPIPS ↓    | CLIP ↑    | DINO ↑    | SSCD ↑    | CLIP-I ↑   | PSNR ↑    | SSIM ↑    |
> > > > |----------------------------------------------|------------|-----------|-----------|-----------|------------|-----------|-----------|
> > > > | **HawkI-Syn (30°, 30°) w/ regularization**   | **0.5661** | **29.9563** | **0.4314** | 0.3638    | **0.8317** | **11.0664** | **0.3162** |
> > > > | HawkI-Syn (30°, 30°) w/o regularization      | 0.5867     | 28.5417    | 0.4122    | **0.3640** | 0.8243     | 10.8272    | 0.2954     |
> > > > | **HawkI-Real (30°, 30°) w/ regularization**  | **0.6201** | **29.8850** | 0.3346    | **0.2588** | 0.8152     | **9.4009** | **0.2184** |
> > > > | HawkI-Real (30°, 30°) w/o regularization     | 0.6257     | 29.0798    | **0.3357** | 0.2401    | **0.8231** | 9.1957     | 0.2014     |
> > > > | **HawkI-Syn (30°, 270°) w/ regularization**  | **0.5744** | **29.1800** | **0.4148** | **0.3684** | **0.8327** | **11.0661** | **0.3047** |
> > > > | HawkI-Syn (30°, 270°) w/o regularization     | 0.5952     | 28.9866    | 0.4098    | 0.3350    | 0.8248     | 10.8656    | 0.2850     |
> > > > | **HawkI-Real (30°, 270°) w/ regularization** | **0.5868** | **30.5489** | **0.4126** | **0.3424** | **0.8708** | **10.6177** | **0.2687** |
> > > > | HawkI-Real (30°, 270°) w/o regularization    | 0.6114     | 29.9184    | 0.4003    | 0.3075    | 0.8541     | 10.2958    | 0.2615     |
> > > > | **HawkI-Syn (-20°, 210°) w/ regularization** | **0.5740** | **29.1144** | **0.4277** | 0.3529    | **0.8280** | **10.9697** | **0.2837** |
> > > > | HawkI-Syn (-20°, 210°) w/o regularization    | 0.5860     | 28.6385    | 0.3969    | **0.3559** | 0.8171     | 10.8401    | 0.2792     |
> > > > | **HawkI-Real (-20°, 210°) w/ regularization**| **0.6185** | **30.6729** | 0.3610    | **0.2880** | **0.8448** | **10.3130** | **0.2223** |
> > > > | HawkI-Real (-20°, 210°) w/o regularization   | 0.6338     | 29.1693    | **0.3817** | 0.2605    | 0.8263     | 10.0794    | 0.2117     |
> > > > | **HawkI-Syn (-20°, 330°) w/ regularization** | **0.5624** | **29.2144** | **0.4487** | **0.3892** | **0.8559** | **11.2175** | **0.3048** |
> > > > | HawkI-Syn (-20°, 330°) w/o regularization    | 0.5714     | 28.4089    | 0.4476    | 0.3870    | 0.8492     | 11.0409    | 0.2947     |
> > > > | **HawkI-Real (-20°, 330°) w/ regularization**| 0.5925     | **29.5090** | **0.3899** | **0.3127** | **0.8689** | **10.6183** | **0.2971** |
> > > > | HawkI-Real (-20°, 330°) w/o regularization   | **0.5894** | 28.8531    | 0.3704    | 0.2954    | 0.8506     | 10.5213    | 0.2828     |
> > > >
> > > > > Table 2: Quantitative Results of Ablation Study. Evaluation of seven metrics demonstrates the superior results of the regularized method over the non-regularized one.
> > > >
> > > > The quantitative and qualitative evaluations demonstrate that applying the **regularization loss term is an effective solution for maintaining consistency.**
> > > >
> > > > Additionally, as **highlighted in the initial Strengths section that BRTu, uyxj, hAuB wrote,** we kindly ask you to consider the fact that **this model is fundamentally a 3D-free method and operates under the constraint of no additional training.** Most novel view synthesis papers **rely on training with 3D datasets**, whereas **our approach does not.**
> > > >
> > > > It is within these constraints that we introduced the regularization loss term to address viewpoint consistency. We would greatly appreciate it if you could reconsider your evaluation, taking into account the strong results achieved **without training using 3D datasets.**
> > > >
> > > > **Despite these constraints,** the consistency of Figures 5 and 6 has been maintained. Specifically, in the case of the pyramid, unlike the existing Zero123++, HawkI, and StableZero123 models, the position of the dark side that should be reflected at a specific angle was accurately captured. For the waterfall, the position of the water stream; for the house image, details such as the chimney and the number of windows for each image were accurately reflected. As for Figure 6, as explained in the main text, the position of the Seine River in the Eiffel Tower scene and the Hawaii beach described above were accurately reflected. Even in the case of the cat photo, the model preserved its form without distortion, accurately reflecting the fur color and the position of the cat’s legs. Please keep in mind that all the results were generated **despite the poor quality of the input images for the 3D-prior model, Zero123++, and evaluate the outcomes accordingly.**
> > > >
> > > > Furthermore, let me address the evaluation metrics. While we incorporated PSNR and SSIM metrics based on your suggestion, we believe that the other metric, **LPIPS is more significant.** This is because PSNR and SSIM are merely metrics for measuring the quality of simple images, whereas metrics like LPIPS can usually used for the NVS task.

---

> ### Author Response · Authors · 2024-11-25
>
> Dear Reviewer,
>
> We would like to sincerely thank you once again for the time and effort you have devoted to reviewing our paper. With the discussion period ending in two days, we would greatly appreciate it if you could review our rebuttal at your convenience, should your schedule permit!
>
> If you have any questions, require additional explanations, or see any areas that need improvement, please feel free to let us know, and we will address them immediately!
>
> We are truly grateful for the valuable feedback you have provided on our research. Additionally, considering the improved methodology and results in the current version, we would be sincerely thankful if you could consider raising your review rating!!!
>
> Hope you have a vibrant start to the week! Thank you very much!!!
>
> With warm regards,
> The Authors

---

> ### Author Response · Authors · 2024-11-27
> **Official Comment by Authors (3)**
>
> You can see which metrics were commonly used in previous studies. For instance, Magic123 [1] uses CLIP and LPIPS for evaluation, Zero-1-to-3 [2] employs LPIPS, ZeroNVS [3] also incorporates LPIPS, and Zero123++ [4] exclusively uses LPIPS. Therefore, I strongly believe that LPIPS should be used when comparing model performance.
>
> According to the LPIPS metric, the performance gap between our model and the model with the lowest performance is 0.1712 (0.7452 - 0.5740, in the HawkI-Syn $(-20^\circ, 210^\circ)$ settings), which is **approximately 5.187 times greater than the 0.033 gap for Zero123++** (refer to Table 1, 0.210 (Zero-1-to-3) - 0.177 (Zero123++ (Ours))). **We have incorporated this feedback and made some revisions to the introduction.** We kindly ask you to review the changes.
>
> Hence, we respectfully request that you reconsider the evaluation using LPIPS, as it is the most widely used evaluation metric.
>
> [1] Qian, Guocheng, et al. "Magic123: One image to high-quality 3d object generation using both 2d and 3d diffusion priors." arXiv preprint arXiv:2306.17843 (2023).
>
> [2] Liu, Ruoshi, et al. "Zero-1-to-3: Zero-shot one image to 3d object." Proceedings of the IEEE/CVF international conference on computer vision. 2023.
>
> [3] Sargent, Kyle, et al. "Zeronvs: Zero-shot 360-degree view synthesis from a single real image." arXiv preprint arXiv:2310.17994 (2023).
>
> [4] Shi, Ruoxi, et al. "Zero123++: a single image to consistent multi-view diffusion base model." arXiv preprint arXiv:2310.15110 (2023).
>
> > **2. Oversaturation issues**
>
> Oversaturation issues can occur in Stable Diffusion. Thank you for pointing this out. The results in this paper were generated using Stable Diffusion 2.1, which employs classifier-free guidance. A paper uploaded to arXiv (https://arxiv.org/abs/2410.02416) in October 2024, titled 'Eliminating Oversaturation and Artifacts of High Guidance Scales in Diffusion Models' [5], proposed a new methodology to address this issue. Once the code is officially released, **we will consider applying this methodology as well.**
>
> [5] Sadat, Seyedmorteza, Otmar Hilliges, and Romann M. Weber. "Eliminating Oversaturation and Artifacts of High Guidance Scales in Diffusion Models." arXiv preprint arXiv:2410.02416 (2024).
>
> > **4. Application limitations**
>
> We are well aware of the concern that Zero123++ can only generate images at six fixed camera poses. Since our guidance image input is provided in the form of images, it is true that applying an excellent 3D-prior model other than Zero123++ could be even more beneficial. We want to clarify that the 3D prior used in our proposed idea is not limited to Zero123++.
>
> We had some practical reasons to use Zero123++ to demonstrate our approach, is clearly stated in Global Rebuttal A3. (This is a question that you did not mention in your initial review. However, other reviewers raised this point, so we included it in the Global Rebuttal.)
>
> For example, the comparison model ZeroNVS **requires 7,500 seconds** to generate a camera-controlled image, and the Stable Zero123 model **takes 1,278 seconds** to generate an image. As more general and faster models come up in the future, our approach can be combined with them. **We mentioned in this detail in the conclusion part.**
>
> > **5. Equation and Writing**
>
> The issue you mentioned with the '?' marks was caused by duplicate references, and we have resolved this error.
>
> Additionally, to avoid confusion between plain text and mathematical notation, **we have revised all equations to use mathematical notation. **
>
> We have corrected **all the errors you pointed out and have uploaded a new version** accordingly.
>
> Furthermore, after acceptance, the **final version will undergo proofreading by a professional writer.**
>
> > **Closing Remarks**
>
> Your primary concerns seem to focus on the results and evaluation, and we sincerely appreciate you bringing these points to our attention. Once this paper is accepted as a camera-ready paper, we will consider further improvements to these aspects.
>
> We believe that perspectives on the results may differ among individuals. However, we would like to **note that reviewers BRTu, uyxj, and hAuB have previously provided positive feedback on the results.**
>
> Additionally, following the rebuttal, a reviewer who had initially given a score of 6 (uyxj) **increased it to 8**, and reviewer hAuB also stated that the rating for **this paper should be raised from 6 to 7.** In light of these points, **we respectfully request that you reconsider your review rating.**
>
> We deeply appreciate the time and effort you have devoted to reviewing our work. While we respect your opinion and acknowledge that everyone may have different perspectives, **we kindly ask for your consideration, as the majority of the reviewers have encouraged this work.**
>
> With warm regards,
> The Authors

---

> > ### Author Response · Authors · 2024-11-28
> >
> > Dear Reviewer yRXV,
> >
> > Could you confirm **if you’ve reviewed our response and whether it or the subsequent discussions have influenced your evaluation?**
> >
> > Thank you for taking the time to review our work; your insights have been incredibly helpful to our research. It is **truly meaningful for us to have someone like you review our paper.**
> >
> > We sincerely appreciate your encouragement of our research potential, **and we would be deeply grateful if you could consider raising the review rating.**
> >
> > Sincerely,
> > The Authors

---

> > > ### Author Response · Authors · 2024-11-30
> > > **Official Comment by Authors: Discussion Period Ending Soon**
> > >
> > > Dear Reviewer yRXV,
> > >
> > > We greatly appreciate your time and effort in reviewing our work and providing detailed feedback. Your insights have been invaluable in helping us refine our paper. As the discussion period is coming to a close in a few days, we kindly request you review our second-round rebuttal at your earliest convenience.
> > >
> > > Thank you once again for your thoughtful consideration.
> > >
> > > With warm regards, The Authors

---

> > > > ### Comment · Area_Chair_5oV2 · 2024-12-01
> > > > **Please continue the discussion**
> > > >
> > > > Dear Reviewer,
> > > >
> > > > Thank you for your prior response, but please continue the discussion. It is an important part of the review process.
> > > >
> > > > Thank, Your AC

---

> ### Author Response · Authors · 2024-12-02
> **Author Rebuttal - Image Consistency [NEW]**
>
> Dear Reviewer yRXV,
>
> Since multiple reviewers have mentioned this issue, **we would like to post an additional rebuttal regarding image consistency.**
>
> As shown in the results attached at the anonymous URL below, we demonstrate that **consistency can be improved by utilizing the Runway Gen-3 Alpha Turbo model as a 3D-prior model** instead of Zero123++.
>
> **URL:** https://github.com/2024nvs/2024nvs
>
> Please kindly note that the core idea of this work is **not the improvement of consistency itself** but rather **the ability to achieve camera control by combining 3D priors with a 3D-free architecture**, which is the novel insight this work has introduced.
>
> We hope that **minor artifacts due to our initial choices in implementation using Zero123++ should be alleviated**, after we demonstrate the new results on using Runway Gen-3 Alpha Turbo as 3D prior.
>
> We kindly request you to reconsider your rating, given our new implementation and results using Runway Gen-3 Alpha Turbo.
>
> With warm regards,
> The Authors

---

> > ### Author Response · Authors · 2024-12-03
> > **Author Rebuttal Reminder - Image Consistency [NEW]**
> >
> > Dear Reviewer yRXV,
> >
> > We would like to remind you about the rebuttal on Image Consistency, as there is not much **time left until the discussion period ends.**
> >
> > Since multiple reviewers have mentioned this issue, **we would like to post an additional rebuttal regarding image consistency.**
> >
> > As shown in the results attached at the anonymous URL below, we demonstrate that **consistency can be improved by utilizing the Runway Gen-3 Alpha Turbo model as a 3D-prior model** instead of Zero123++.
> >
> > **URL:** https://github.com/2024nvs/2024nvs
> >
> > Please kindly note that the core idea of this work is **not the improvement of consistency itself** but rather **the ability to achieve camera control by combining 3D priors with a 3D-free architecture**, which is the novel insight this work has introduced.
> >
> > We hope that **minor artifacts due to our initial choices in implementation using Zero123++ should be alleviated**, after we demonstrate the new results on using Runway Gen-3 Alpha Turbo as 3D prior.
> >
> > We kindly request you to reconsider your rating, given our new implementation and results using Runway Gen-3 Alpha Turbo.
> >
> > With warm regards,
> > The Authors

---

### Official Review · Reviewer_hAuB · 2024-10-24

**Soundness:** 3
**Presentation:** 3
**Contribution:** 3
**Rating:** 6
**Confidence:** 3

**Summary:**

The paper tackles the problem of novel view synthesis from a single image input of a scene by using textual camera view description. The problem statement is very similar to previous works Zero123++, HawkI, except that an additional guidance image is introduced, obtained through a pre-trained model Zero123++, tackling the same problem. The guidance image network Zero123++ is trained on 3D shapes, but for a single object. The introduced method uses a series of 2 double-step optimizations. The first stage optimizes the diffusion-enriched CLIP space of the input view and the second stage does the same for the target view. The results show that the proposed method has better performance in comparison to previous works. It tends to be more consistent in showing both central object and background from a variety of viewpoints (which are specified in terms of azimuth and elevation angles).

**Strengths:**

1. The method shows a good performance of novel view synthesis on a single view of a scene with object + background. As far as related works are concerned, the method shows - on average - better results using a textual prompt and without depth information for object + background when novel views are synthesized.

2. The results show a consistent performance gain over previous works in both real and synthetically generated data.

**Weaknesses:**

1. The method is quite convoluted with 4 different test-time optimization steps. No time comparisons are provided. It would be important to mention typical in inference times...

2. Looking into the intermediate and final results, as well as the method, the main task of aligning the camera appears to be the result of the Zero123++ method, while the test-time optimizations generally improves the image quality. This is also reflected in the particular choice of angles, as only those angles are trained for in Zero123++. However, for small angles (30,30), often the test-time optimization worsens the results, e.g., figure 6, angles = (30,30) for cats. Same for figure 5, angles = (30,30)- the house changes completely after optimization. Due to this reason, it may be a bit dubious to stipulate object + background novel view synthesis as contribution. After the evaluations, both the contributions (1) & (2) could be attributed - at least in part - to Zero123++.

3. The architecture diagram in Figure 2 is not complete in the info provided. Missing: which losses are used? These should be made clear together with the equations (1,2) since this is the crux of the proposed method.

4. L215--L216- "Our model eliminates the need for training data, 3D data or multi-view data". This is somewhat misleading. The camera alignment largely appears to be performed by the guidance method Zero123++, which is trained on Objaverse and more importantly on  the same task. The method implicitly uses Objaverse via Zero123++, a competing method used in the proposed method. It is not clear how Zero123++ performs test-time optimization without the 3D data.

5. The paper glosses over some of the limitations in the results, like the house not being shown in a very consistent fashion when viewpoint is changed. Probably there is a difference in performance between renowned monuments with many photographs that can be scraped from the Internet, compared to rather unique objects / situations.

6. Given the complete approach, the novelty of the work mainly consists of the combination of methods, whereas the actual novelty drags in the cost of extensive inference optimization.

7. The English could benefit from some slight polishing at times

**Questions:**

It would be good if the authors clearly define what they mean by azimuth and elevation angles. The typical definition of the latter would be relative to the horizontal plane. Is that the case here as well, or are angles relative to the guidance view ? In any case, the angles specified in the image captions do not seem to follow one consistent set of definitions... Elevation angles are taking up values larger than 90 degrees, for instance.

Related work that could be mentioned: Depth self-supervision for single image novel view synthesis, 2023 Giovanni Minelli et al.

The creation of videos with continuously changing viewpoints would be telling when it comes to consistency and angle correctness...

---

> ### Author Response · Authors · 2024-11-22
> **Rebuttal by Authors**
>
> Thank you sincerely for your constructive and detailed feedback. We fully understand the points you have raised and have provided responses below.
>
> > **(1) The method is quite convoluted with 4 different test-time optimization steps. No time comparisons are provided. It would be important to mention typical in inference times.**
>
> Please refer to A1 of the global rebuttal at the top!
>
> > **(2) Looking into the intermediate and final results, as well as the method, the main task of aligning the camera appears to be the result of the Zero123++ method, while the test-time optimizations generally improves the image quality. This is also reflected in the particular choice of angles, as only those angles are trained for in Zero123++. However, for small angles (30,30), often the test-time optimization worsens the results, e.g., figure 6, angles = (30,30) for cats. Same for figure 5, angles = (30,30)- the house changes completely after optimization. Due to this reason, it may be a bit dubious to stipulate object + background novel view synthesis as contribution. After the evaluations, both the contributions (1) & (2) could be attributed - at least in part - to Zero123++.**
>
> Please refer to A2 of the global rebuttal at the top!
>
> > **(3) The architecture diagram in Figure 2 is not complete in the info provided. Missing: which losses are used? These should be made clear together with the equations (1,2) since this is the crux of the proposed method.**
>
> > **(7) The English could benefit from some slight polishing at times**
>
> > **(8) It would be good if the authors clearly define what they mean by azimuth and elevation angles. The typical definition of the latter would be relative to the horizontal plane. Is that the case here as well, or are angles relative to the guidance view ? In any case, the angles specified in the image captions do not seem to follow one consistent set of definitions... Elevation angles are taking up values larger than 90 degrees, for instance. Related work that could be mentioned: Depth self-supervision for single image novel view synthesis, 2023 Giovanni Minelli et al.**
>
> We have made the requested revisions and improved the clarity of the four steps used in our paper.
>
> Additionally, we have updated the method figure to clearly indicate which losses are used, making it easier to understand.
>
> Moreover, please note that improvements have also been made to the writing of the paper.
>
> Furthermore, as you suggested, we have clarified the definitions of azimuth and elevation. We use the azimuth and elevation as defined by Zero123++.
>
> > **(4) L215--L216- "Our model eliminates the need for training data, 3D data or multi-view data". This is somewhat misleading. The camera alignment largely appears to be performed by the guidance method Zero123++, which is trained on Objaverse and more importantly on the same task. The method implicitly uses Objaverse via Zero123++, a competing method used in the proposed method. It is not clear how Zero123++ performs test-time optimization without the 3D data.**
>
> That is an excellent point.
>
> To clarify, **what we mean by '3D-free' is that our method does not require any "additional 3D" data for training or generating results beyond the data provided to the pretrained model.**
>
> **Many of the methods** mentioned in our Introduction and Related Works sections **rely on** additional **3D data** to **achieve good performance.**
>
> In contrast, our method **does not require** such **additional 3D data**.
>
> > **(5) The paper glosses over some of the limitations in the results, like the house not being shown in a very consistent fashion when viewpoint is changed. Probably there is a difference in performance between renowned monuments with many photographs that can be scraped from the Internet, compared to rather unique objects / situations.**
>
> Please refer to A2 of the global rebuttal at the top!
>
> > **(6) Given the complete approach, the novelty of the work mainly consists of the combination of methods, whereas the actual novelty drags in the cost of extensive inference optimization.**
>
> Please refer to A1 of the global rebuttal at the top!
>
> > **(9) The creation of videos with continuously changing viewpoints would be telling when it comes to consistency and angle correctness…**
>
> Thank you sincerely for your excellent suggestion. Consistency has been significantly improved through the regularization loss, as demonstrated in the paper. If we have the opportunity to submit a camera-ready version, we will consider ways to further emphasize consistency in the paper.
>
> **Given the improved results/methodology presented in this revision, and your initial assessment of Acceptance (6), we would be deeply grateful if you could raise your review ratings that better reflect your encouraging evaluation!**
>
> Thank you sincerely for taking the time to provide such detailed reviews. If you have any additional questions, please feel free to reach out at any time!

---

> > ### Comment · Area_Chair_5oV2 · 2024-11-24
> > **Discussion Period Ending Soon**
> >
> > Dear Reviewer,
> >
> > The discussion period will end soon. Please take a look at the author's comments and begin a discussion.
> >
> > Thanks, Your AC

---

> > ### Comment · Reviewer_hAuB · 2024-11-25
> >
> > Thanks for taking the time to answer to my questions. The clarifications concerning computation time are highly appreciated.  Although concerns regarding image consistency across views persist, I believe discussions regarding increased user control in generative AI are important and this paper contributes to this strand. I am willing to increase my grading score to 7.

---

> > > ### Author Response · Authors · 2024-11-25
> > >
> > > Dear Reviewer hAuB,
> > >
> > > First, I would like to sincerely thank you for deciding to raise the review rating by one level. It seems that the **previous review rating ("6: marginally above the acceptance threshold") is still being displayed.**
> > >
> > > At your convenience, **we would greatly appreciate it if you could review the rating and update it to reflect the one-level increase** (**"8: accept, good paper"**).
> > >
> > > Once again, thank you for deciding to raise the rating, and we will do our best to address the image consistency you mentioned even after acceptance.
> > >
> > > Thank you very much!
> > >
> > > With warm regards,
> > > The Authors

---

> > > > ### Comment · Reviewer_hAuB · 2024-11-25
> > > >
> > > > I said I would be willing to upgrade to 7, but that grade isn't possible unfortunately. Calling the paper a good paper deserving a solid advise to accept maybe is a bridge too far. I wanted to indicate that I would rather go up with my grading than going down. But the paper still falls short a bit from really delivering what is promised at the beginning. But it is a worthwhile contribution, that I would be interested in seeing in the proceedings.

---

> > > > > ### Author Response · Authors · 2024-11-26
> > > > >
> > > > > Dear Reviewer hAuB,
> > > > >
> > > > > We sincerely thank you once again for your thoughtful feedback and for evaluating our paper positively. We truly appreciate the time and effort you have invested in reviewing our work. You mentioned that you would increase the grading score to 7, but unfortunately, it seems that the ICLR 2025 system does not allow this score to be selected.
> > > > >
> > > > > While the discussion period has been extended, **we now have just over about 24 hours remaining to make any final adjustments to the paper.** May we kindly ask if there are **additional points or aspects, particularly in the experimental section, that require further clarification or elaboration?** We are fully prepared to incorporate your suggestions within the given timeframe to ensure that your concerns are addressed as thoroughly as possible!
> > > > >
> > > > > As you mentioned, your active support is crucial for this paper to be accepted in the proceedings. For example, reviewer uyxj, who initially rated the paper with a 6, **raised their score to 8 after reviewing the rebuttal and the revised paper.** We would greatly appreciate it **if you could do similar consideration!!!**
> > > > >
> > > > > So, we would greatly appreciate it **if you could reconsider raising your review rating!** Since submitting the rebuttal, we have made further improvements to the results, particularly in areas you highlighted as strengths.
> > > > >
> > > > > The refinements further validate our approach, and we have diligently addressed the points you raised, leading to substantial enhancements in the highlighted areas.
> > > > >
> > > > > Given these improvements, the positive feedback we’ve received, and your overall evaluation of 7 points beyond '6: marginally above the acceptance threshold' criteria (with 'Good' ratings in Soundness, Presentation, and Contribution, which is one-level lower than full points 'Excellent'), **we respectfully suggest that a higher score might better reflect your encouraging assessment and the merit of the paper!!!**
> > > > >
> > > > > Thank you very much for your valuable feedback and for considering this request. We are deeply grateful for your support.
> > > > >
> > > > > With warm regards,
> > > > > The Authors

---

> > > > > > ### Author Response · Authors · 2024-11-28
> > > > > >
> > > > > > Dear Reviewer hAuB,
> > > > > >
> > > > > > Could you confirm **if you’ve reviewed our response and whether it or the subsequent discussions have influenced your evaluation?**
> > > > > >
> > > > > > Thank you for taking the time to review our work; your insights have been incredibly helpful to our research. It is **truly meaningful for us to have someone like you review our paper.**
> > > > > >
> > > > > > We sincerely appreciate your encouragement of our research potential, **and we would be deeply grateful if you could consider raising the review rating.**
> > > > > >
> > > > > > Sincerely,
> > > > > > The Authors

---

> > > > > > > ### Author Response · Authors · 2024-11-30
> > > > > > > **Official Comment by Authors: Discussion Period Ending Soon**
> > > > > > >
> > > > > > > Dear Reviewer hAuB,
> > > > > > >
> > > > > > > We greatly appreciate your time and effort in reviewing our work and providing detailed feedback. Your insights have been invaluable in helping us refine our paper. As the discussion period is coming to a close in a few days, we kindly request you review our previous comment at your earliest convenience.
> > > > > > >
> > > > > > > Thank you once again for your thoughtful consideration.
> > > > > > >
> > > > > > > With warm regards, The Authors

---

> > > > > > > > ### Author Response · Authors · 2024-12-02
> > > > > > > > **Author Rebuttal - Image Consistency [NEW]**
> > > > > > > >
> > > > > > > > Dear Reviewer hAuB,
> > > > > > > >
> > > > > > > > Since multiple reviewers have mentioned this issue, **we would like to post an additional rebuttal regarding image consistency.**
> > > > > > > >
> > > > > > > > As shown in the results attached at the anonymous URL below, we demonstrate that **consistency can be improved by utilizing the Runway Gen-3 Alpha Turbo model as a 3D-prior model** instead of Zero123++.
> > > > > > > >
> > > > > > > > **URL:** https://github.com/2024nvs/2024nvs
> > > > > > > >
> > > > > > > > Please kindly note that the core idea of this work is **not the improvement of consistency itself** but rather **the ability to achieve camera control by combining 3D priors with a 3D-free architecture**, which is the novel insight this work has introduced.
> > > > > > > >
> > > > > > > > We hope that **minor artifacts due to our initial choices in implementation using Zero123++ should be alleviated**, after we demonstrate the new results on using Runway Gen-3 Alpha Turbo as 3D prior.
> > > > > > > >
> > > > > > > > We kindly request you to reconsider your rating, given our new implementation and results using Runway Gen-3 Alpha Turbo.
> > > > > > > >
> > > > > > > > With warm regards,
> > > > > > > > The Authors

---

> > > > > > > > > ### Author Response · Authors · 2024-12-03
> > > > > > > > > **Author Rebuttal Reminder - Image Consistency [NEW]**
> > > > > > > > >
> > > > > > > > > Dear Reviewer hAuB,
> > > > > > > > >
> > > > > > > > > We would like to remind you about the rebuttal on Image Consistency, as there is not much **time left until the discussion period ends.**
> > > > > > > > >
> > > > > > > > > Since multiple reviewers have mentioned this issue, **we would like to post an additional rebuttal regarding image consistency.**
> > > > > > > > >
> > > > > > > > > As shown in the results attached at the anonymous URL below, we demonstrate that **consistency can be improved by utilizing the Runway Gen-3 Alpha Turbo model as a 3D-prior model** instead of Zero123++.
> > > > > > > > >
> > > > > > > > > **URL:** https://github.com/2024nvs/2024nvs
> > > > > > > > >
> > > > > > > > > Please kindly note that the core idea of this work is **not the improvement of consistency itself** but rather **the ability to achieve camera control by combining 3D priors with a 3D-free architecture**, which is the novel insight this work has introduced.
> > > > > > > > >
> > > > > > > > > We hope that **minor artifacts due to our initial choices in implementation using Zero123++ should be alleviated**, after we demonstrate the new results on using Runway Gen-3 Alpha Turbo as 3D prior.
> > > > > > > > >
> > > > > > > > > We kindly request you to reconsider your rating, given our new implementation and results using Runway Gen-3 Alpha Turbo.
> > > > > > > > >
> > > > > > > > > With warm regards,
> > > > > > > > > The Authors

---

> ### Author Response · Authors · 2024-11-25
>
> Dear Reviewer,
>
> We would like to sincerely thank you once again for the time and effort you have devoted to reviewing our paper. With the discussion period ending in two days, we would greatly appreciate it if you could review our rebuttal at your convenience, should your schedule permit!
>
> If you have any questions, require additional explanations, or see any areas that need improvement, please feel free to let us know, and we will address them immediately!
>
> We are truly grateful for the valuable feedback you have provided on our research. Additionally, considering the improved methodology and results in the current version, we would be sincerely thankful if you could consider raising your review rating!!!
>
> Hope you have a vibrant start to the week! Thank you very much!!!
>
> With warm regards,
> The Authors

---

### Official Review · Reviewer_9Fq5 · 2024-10-28

**Soundness:** 1
**Presentation:** 1
**Contribution:** 2
**Rating:** 3
**Confidence:** 4

**Summary:**

The paper presents an optimization-based sampling method to generate novel views at inference time. It utilizes multi-view images generated by a pre-trained diffusion model and optimizes the text embeddings and LoRA layers of a text-to-image diffusion model to guide the novel view sampling process.

**Strengths:**

1. The paper investigates an approach that sequentially optimizes the text embeddings and LoRA layers of a diffusion model to guide the novel view synthesis process.

**Weaknesses:**

1. The proposed method relies on a pre-trained multi-view diffusion model, Zero123++, and thus shares its limitations: restricted view generation (Zero123++ generates six fixed-view images around the object), modification of background and object, handling multiple objects, and pose misalignment.
2. The evaluation can be seem unfair. For apple to apple comparison, the paper could compare with running the proposed method on Zero123++. Also, measure the CLIP score seem unfair as the proposed method optimizes the CLIP text embeddings.
3. The proposed method introduces additional computational costs: generating a multi-view guidance images from a pre-trained diffusion model and the optimization of text embeddings to reconstruct $I_{input}$ and $I_{view}$.

* L.86, L.90: Some terms are used in contradicting manner. 3D-free methods like Zero123++  and 3D-based models like Zero123++.
* L.95: we (1) we
* Eqn. 1: The optimization task is provided without details. $L, f, x_t$ is not introduced before.

**Questions:**

The method could be presented with more detail. Specifically, how do Eq. 1 and Eq. 2 work? Does the diffusion model output clean predicted samples? If not, how is the loss computed between the noisy latent $x_t$ and the clean image $I_{input}$?

---

> ### Author Response · Authors · 2024-11-22
> **Rebuttal by Authors**
>
> Thank you sincerely for your constructive and detailed feedback. We fully understand the points you have raised and have provided responses below.
>
> > **(1) The proposed method relies on a pre-trained multi-view diffusion model, Zero123++, and thus shares its limitations: restricted view generation (Zero123++ generates six fixed-view images around the object), modification of background and object, handling multiple objects, and pose misalignment.**
>
> Please refer to A3 of the global rebuttal at the top!
>
> > **(2) The evaluation can be seem unfair. For apple to apple comparison, the paper could compare with running the proposed method on Zero123++. Also, measure the CLIP score seem unfair as the proposed method optimizes the CLIP text embeddings.**
>
> Thank you for mentioning the metrics.
>
> We followed the evaluation approach of the baseline model, HawkI. However, metrics like View-CLIP, CLIPD, and View-CLIPD, while relevant to our task, were deemed somewhat mismatched (and potentially unfair), so we decided to exclude them.
>
> Instead, we added metrics commonly used in other view synthesis tasks, such as **PSNR and SSIM, as well as DINO and SSCD**, to enhance the reliability of our evaluation. The **LPIPS metric used by Zero123++** has already been included in the previous paper we submitted.
>
> > **(3) The proposed method introduces additional computational costs: generating a multi-view guidance images from a pre-trained diffusion model and the optimization of text embeddings to reconstruct $I_{input}$ and $I_{view}$**
>
> Please refer to A1 of the global rebuttal at the top!
>
> > **(4) L.86, L.90: Some terms are used in contradicting manner. 3D-free methods like Zero123++ and 3D-based models like Zero123++. / L.95: we (1) we / Eqn. 1: The optimization task is provided without details. $L, f, x_{t}$ is not introduced before.**
>
> > **(5) The method could be presented with more detail. Specifically, how do Eq. 1 and Eq. 2 work? Does the diffusion model output clean predicted samples? If not, how is the loss computed between the noisy latent $x_{t}$ and the clean image $I_{input}$?**
>
> We have reviewed and revised the writing as you suggested, and added more detailed information about the variables $L, f, x_{t}$ improving the overall quality of the paper.
>
> In addition, as another reviewer pointed out that the method was somewhat difficult to understand, we have explained all the loss functions used in the four-step process and improved the method figure by adding numbered labels for better clarity.
>
> There have been **significant improvements in the results and the paper**, and based on the **positive evaluations from several reviewers regarding the novelty and potential of our idea**, we would like to kindly and respectfully ask if you could consider raising the review rating for our paper.
>
> Thank you sincerely for taking the time to provide such detailed reviews. If you have any additional questions, please feel free to reach out at any time!

---

> > ### Comment · Area_Chair_5oV2 · 2024-11-24
> > **Discussion Period Ending Soon**
> >
> > Dear Reviewer,
> >
> > The discussion period will end soon. Please take a look at the author's comments and begin a discussion.
> >
> > Thanks, Your AC

---

> ### Author Response · Authors · 2024-11-25
>
> Dear Reviewer,
>
> We would like to sincerely thank you once again for the time and effort you have devoted to reviewing our paper. With the discussion period ending in two days, we would greatly appreciate it if you could review our rebuttal at your convenience, should your schedule permit!
>
> If you have any questions, require additional explanations, or see any areas that need improvement, please feel free to let us know, and we will address them immediately!
>
> We are truly grateful for the valuable feedback you have provided on our research. Additionally, considering the improved methodology and results in the current version, we would be sincerely thankful if you could consider raising your review rating!!!
>
> Hope you have a vibrant start to the week! Thank you very much!!!
>
> With warm regards,
> The Authors

---

> ### Comment · Reviewer_9Fq5 · 2024-11-26
>
> I appreciate the authors for clarifications on evaluation metrics and optimization tasks.
>
> Still, the applicability of the proposed method remains unclear to me due to the dependency of Zero123++ which I list as follows.
> 1. Zero123++ can only generate images at **six fixed** camera poses which means the proposed method can also be limited in generating these 6 poses. This is also reflected in the result section, where the evaluation is performed only using a discrete set of views. This limitation may be critical as in most NVS tasks, users may want to freely move around the object/scene.
>
> 2. While the outputs from Zero123++ generate quite consistent images, it **does not guarantee the preservation of the source view image**. This is also reflected in Figs. 5-6 where the generated outputs of the proposed method alter the content of the source view image.
>
> 3. Zero123++ may **produce frames with incorrect pose information**. While the model is capable of generating plausible target view images, it does not incorporate explicit physical constraints, which can result in inaccuracies when generating images with precise pose information.
>
> While I do not claim that the use of Zero123++ is flawed, I believe the paper should explain how these issues are addressed in their proposed method and how adopting the multi-view diffusion model can outperform inpainting-based methods or pose-conditioned video generative models? This leads to additional concern as the authors does not present comparisons to other types of methods.
>
> Additionally, the optimization tasks seem like to introduce a lot of burdens as it accumulates the gradient of all timesteps (Eqs. 1-3). It raises questions on how the gradients are backpropagated.
>
> Once again, I appreciate the authors for providing clarifications. It would be great if the authors can clarify the concerns raised above.

---

> ### Author Response · Authors · 2024-11-27
> **Official Comment by Authors (1)**
>
> Dear Reviewer 9Fq5.
>
> Thank you sincerely for providing insightful comments and making such valuable points.
>
> > **1. Zero123++ can only generate images at six fixed camera poses which means the proposed method can also be limited in generating these 6 poses. This is also reflected in the result section, where the evaluation is performed only using a discrete set of views. This limitation may be critical as in most NVS tasks, users may want to freely move around the object/scene.**
>
> We are well aware of the concern that Zero123++ can only generate images at six fixed camera poses.
>
> Since our guidance image input is provided in the form of images, it is true that applying an excellent 3D-prior model other than Zero123++ could be even more beneficial. We want to clarify that the 3D prior used in our proposed idea **is not limited to Zero123++.**
>
> We had some **practical reasons** to use Zero123++ to demonstrate our approach, is clearly stated in Global Rebuttal A3.
>
> For example, the comparison model ZeroNVS **requires 7,500 seconds** to generate a camera-controlled image, and the Stable Zero123 model **takes 1,278 seconds** to generate an image.
>
> As more general and faster models come up in the future, our approach can be combined with them. **We mentioned in this detail in the conclusion part.**
>
> > **2. While the outputs from Zero123++ generate quite consistent images, it does not guarantee the preservation of the source view image. This is also reflected in Figs. 5-6 where the generated outputs of the proposed method alter the content of the source view image.**
>
> > **3. Zero123++ may produce frames with incorrect pose information. While the model is capable of generating plausible target view images, it does not incorporate explicit physical constraints, which can result in inaccuracies when generating images with precise pose information.**
>
> We are well aware of the concern that Zero123++ may fail to preserve the content of the source view image or provide incorrect pose information, and this has been explicitly stated in Section 5.1.
>
> To address this issue, we applied a regularization loss to resolve it. Accordingly, as indirectly mentioned in Section 4.1.5, we clarified that the 3D-prior model performs **weak or pseudo guidance.** This explanation is identical to the response provided to Reviewer BRTu. Please refer to the details and please check Section 5.3, Figure 7, and Table 2.
>
> In particular, the examples highlighted in the ablation study shown in Figure 7 include scenes of a Hawaiian beach and a waterfall. In both cases, the Zero123++ model failed to provide accurate viewpoints and guidance to the model. However, the application of the regularization loss **significantly improved the consistency of the textures in the Hawaiian beach scene and enhanced the consistency of the rock textures generated in the waterfall scene.**
>
> We kindly ask you to refer to Section 5.3 for more detailed information on this aspect.
>
> > *Part of sentences from Section 5.3*: "For the Hawaiian beach scene, the **output generated with the regularization term exhibits a more consistent style** compared to the output produced without it. Despite the inaccuracies in the Zero123++ guidance images, the regularization term facilitates **more reliable camera control than the results generated without it.**
> Similarly, in the waterfall scene, the regularization term enhances the consistency of the generated rock textures surrounding the waterfall. **Without the regularization term, these textures are inconsistently represented; however, with it, the style is maintained more faithfully.** Once again, Zero123++ does not provide accurate guidance in this case, underscoring the significant contribution of the regularization loss to improved control and visual coherence in the generated images."
>
> We also conducted an **ablation study** to compare the results with and without the application of the regularization loss. By analyzing seven image evaluation metrics presented in the existing evaluation — LPIPS, CLIP, DINO, SSCD, CLIP-I, PSNR, and SSIM — we demonstrated superior performance, thereby providing quantitative evaluations. We have attached the results in a table below for your reference, and we would appreciate it if you could take a look.

---

> > ### Author Response · Authors · 2024-11-27
> > **Official Comment by Authors (2)**
> >
> > > | Dataset, Angle, Method                       | LPIPS ↓    | CLIP ↑    | DINO ↑    | SSCD ↑    | CLIP-I ↑   | PSNR ↑    | SSIM ↑    |
> > |----------------------------------------------|------------|-----------|-----------|-----------|------------|-----------|-----------|
> > | **HawkI-Syn (30°, 30°) w/ regularization**   | **0.5661** | **29.9563** | **0.4314** | 0.3638    | **0.8317** | **11.0664** | **0.3162** |
> > | HawkI-Syn (30°, 30°) w/o regularization      | 0.5867     | 28.5417    | 0.4122    | **0.3640** | 0.8243     | 10.8272    | 0.2954     |
> > | **HawkI-Real (30°, 30°) w/ regularization**  | **0.6201** | **29.8850** | 0.3346    | **0.2588** | 0.8152     | **9.4009** | **0.2184** |
> > | HawkI-Real (30°, 30°) w/o regularization     | 0.6257     | 29.0798    | **0.3357** | 0.2401    | **0.8231** | 9.1957     | 0.2014     |
> > | **HawkI-Syn (30°, 270°) w/ regularization**  | **0.5744** | **29.1800** | **0.4148** | **0.3684** | **0.8327** | **11.0661** | **0.3047** |
> > | HawkI-Syn (30°, 270°) w/o regularization     | 0.5952     | 28.9866    | 0.4098    | 0.3350    | 0.8248     | 10.8656    | 0.2850     |
> > | **HawkI-Real (30°, 270°) w/ regularization** | **0.5868** | **30.5489** | **0.4126** | **0.3424** | **0.8708** | **10.6177** | **0.2687** |
> > | HawkI-Real (30°, 270°) w/o regularization    | 0.6114     | 29.9184    | 0.4003    | 0.3075    | 0.8541     | 10.2958    | 0.2615     |
> > | **HawkI-Syn (-20°, 210°) w/ regularization** | **0.5740** | **29.1144** | **0.4277** | 0.3529    | **0.8280** | **10.9697** | **0.2837** |
> > | HawkI-Syn (-20°, 210°) w/o regularization    | 0.5860     | 28.6385    | 0.3969    | **0.3559** | 0.8171     | 10.8401    | 0.2792     |
> > | **HawkI-Real (-20°, 210°) w/ regularization**| **0.6185** | **30.6729** | 0.3610    | **0.2880** | **0.8448** | **10.3130** | **0.2223** |
> > | HawkI-Real (-20°, 210°) w/o regularization   | 0.6338     | 29.1693    | **0.3817** | 0.2605    | 0.8263     | 10.0794    | 0.2117     |
> > | **HawkI-Syn (-20°, 330°) w/ regularization** | **0.5624** | **29.2144** | **0.4487** | **0.3892** | **0.8559** | **11.2175** | **0.3048** |
> > | HawkI-Syn (-20°, 330°) w/o regularization    | 0.5714     | 28.4089    | 0.4476    | 0.3870    | 0.8492     | 11.0409    | 0.2947     |
> > | **HawkI-Real (-20°, 330°) w/ regularization**| 0.5925     | **29.5090** | **0.3899** | **0.3127** | **0.8689** | **10.6183** | **0.2971** |
> > | HawkI-Real (-20°, 330°) w/o regularization   | **0.5894** | 28.8531    | 0.3704    | 0.2954    | 0.8506     | 10.5213    | 0.2828     |
> >
> > > Table 2: Quantitative Results of Ablation Study. Evaluation of seven metrics demonstrates the superior results of the regularized method over the non-regularized one.
> >
> > The quantitative and qualitative evaluations demonstrate that applying the **regularization loss term is an effective solution for maintaining consistency.**
> >
> > Additionally, as **highlighted in the initial Strengths section that BRTu, uyxj, hAuB wrote,** we kindly ask you to consider the fact that **this model is fundamentally a 3D-free method and operates under the constraint of no additional training.** Most novel view synthesis papers **rely on training with 3D datasets**, whereas **our approach does not.**
> >
> > It is within these constraints that we introduced the regularization loss term to address viewpoint consistency. We would greatly appreciate it if you could reconsider your evaluation, taking into account the strong results achieved **without training using 3D datasets.**
> >
> > **Despite these constraints,** the consistency of Figures 5 and 6 has been maintained. Specifically, in the case of the pyramid, unlike the existing Zero123++, HawkI, and StableZero123 models, the position of the dark side that should be reflected at a specific angle was accurately captured. For the waterfall, the position of the water stream; for the house image, details such as the chimney and the number of windows for each image were accurately reflected. As for Figure 6, as explained in the main text, the position of the Seine River in the Eiffel Tower scene and the Hawaii beach described above were accurately reflected. Even in the case of the cat photo, the model preserved its form without distortion, accurately reflecting the fur color and the position of the cat’s legs. Please keep in mind that all the results were generated **despite the poor quality of the input images for the 3D-prior model, Zero123++, and evaluate the outcomes accordingly.**

---

> ### Author Response · Authors · 2024-11-27
> **Official Comment by Authors (3)**
>
> > **4. While I do not claim that the use of Zero123++ is flawed, I believe the paper should explain how these issues are addressed in their proposed method and how adopting the multi-view diffusion model can outperform inpainting-based methods or pose-conditioned video generative models? This leads to additional concern as the authors does not present comparisons to other types of methods.**
>
> Thank you for the insightful comment. However, inpainting-based methods or pose-conditioned video generative models differ significantly from our method, making direct comparisons challenging.
>
> First, regarding pose-conditioned video generative models, a representative example would be Pix2Pix3D [1] (https://arxiv.org/abs/2302.08509). Our method operates in an environment where an input image and text are given to perform camera control, whereas Pix2Pix3D uses a 2D label map as input, **which inherently defines a different setting.**
>
> Moreover, our approach is not limited to simple environments such as faces or objects **but tackles a more complex task** by performing camera control on **synthetic data with backgrounds or real-world data with backgrounds.**
>
> As for inpainting-based methods, we can take the latest study, MVInpainter [2] (https://arxiv.org/pdf/2408.08000), as an example. Since this study uses 3D data, **it is not 3D-free, and therefore, it cannot serve as a direct comparison to our method.**
>
> [1] Deng, Kangle, et al. "3d-aware conditional image synthesis." Proceedings of the IEEE/CVF Conference on Computer Vision and Pattern Recognition. 2023.
>
> [2] Cao, Chenjie, et al. "Mvinpainter: Learning multi-view consistent inpainting to bridge 2d and 3d editing." arXiv preprint arXiv:2408.08000 (2024).
>
> > **5. Additionally, the optimization tasks seem like to introduce a lot of burdens as it accumulates the gradient of all timesteps (Eqs. 1-3). It raises questions on how the gradients are backpropagated.**
>
> We will explain this part based on our implemented code.
>
> Our implementation uses configurable `gradient_accumulation_steps`, which allows for flexible control over how gradients are accumulated. This ensures that the computational overhead can be managed by balancing memory and performance considerations.
>
> Additionally, the use of `accelerator.backward()` optimizes gradient computation and synchronization across devices in distributed setups.
>
> Gradients are only computed for parameters explicitly involved in the optimization task (e.g., `text_embeddings`), while other parts of the model are frozen using `requires_grad_(False)` to minimize unnecessary computations.
>
> To reduce computational costs, we employ techniques such as gradient checkpointing, selective accumulation, and efficient memory utilization through frameworks like `accelerator`. These techniques ensure that the optimization process remains computationally tractable without compromising model performance.
>
> > **Closing remarks**
>
> Your primary concerns seem to focus on the applicability of Zero123++, and we sincerely appreciate you bringing these points to our attention. Once this paper is accepted as a camera-ready paper, we will consider further improvements to these aspects.
>
> We believe that perspectives on the results may differ among individuals. However, we would like to **note that reviewers BRTu, uyxj, and hAuB have previously provided positive feedback on the results.**
>
> Additionally, following the rebuttal, a reviewer who had initially given a score of 6 (uyxj) **increased it to 8**, and reviewer hAuB also stated that the rating for **this paper should be raised from 6 to 7.** In light of these points, **we respectfully request that you reconsider your review rating.**
>
> We deeply appreciate the time and effort you have devoted to reviewing our work. While we respect your opinion and acknowledge that everyone may have different perspectives, **we kindly ask for your consideration, as the majority of the reviewers have encouraged this work.**
>
> With warm regards,
> The Authors

---

> > ### Author Response · Authors · 2024-11-28
> >
> > Dear Reviewer 9Fq5,
> >
> > Could you confirm **if you’ve reviewed our response and whether it or the subsequent discussions have influenced your evaluation?**
> >
> > Thank you for taking the time to review our work; your insights have been incredibly helpful to our research. It is **truly meaningful for us to have someone like you review our paper.**
> >
> > We sincerely appreciate your encouragement of our research potential, **and we would be deeply grateful if you could consider raising the review rating.**
> >
> > Sincerely,
> > The Authors

---

> ### Author Response · Authors · 2024-11-30
> **Official Comment by Authors: Discussion Period Ending Soon**
>
> Dear Reviewer 9Fq5,
>
> We greatly appreciate your time and effort in reviewing our work and providing detailed feedback. Your insights have been invaluable in helping us refine our paper. As the discussion period is coming to a close in a few days, we kindly request you review our second-round rebuttal at your earliest convenience.
>
> Thank you once again for your thoughtful consideration.
>
> With warm regards, The Authors

---

> > ### Comment · Area_Chair_5oV2 · 2024-12-01
> > **Discuss**
> >
> > Dear Reviewer,
> >
> > Discussion is a critical part of the review process. Please engage with the authors.
> >
> > Thanks, Your AC

---

> ### Author Response · Authors · 2024-12-02
> **Author Rebuttal - Image Consistency [NEW]**
>
> Dear Reviewer 9Fq5,
>
> Since multiple reviewers have mentioned this issue, **we would like to post an additional rebuttal regarding image consistency.**
>
> As shown in the results attached at the anonymous URL below, we demonstrate that **consistency can be improved by utilizing the Runway Gen-3 Alpha Turbo model as a 3D-prior model** instead of Zero123++.
>
> **URL:** https://github.com/2024nvs/2024nvs
>
> Please kindly note that the core idea of this work is **not the improvement of consistency itself** but rather **the ability to achieve camera control by combining 3D priors with a 3D-free architecture**, which is the novel insight this work has introduced.
>
> We hope that **minor artifacts due to our initial choices in implementation using Zero123++ should be alleviated**, after we demonstrate the new results on using Runway Gen-3 Alpha Turbo as 3D prior.
>
> We kindly request you to reconsider your rating, given our new implementation and results using Runway Gen-3 Alpha Turbo.
>
> With warm regards,
> The Authors

---

> > ### Author Response · Authors · 2024-12-03
> > **Author Rebuttal Reminder - Image Consistency [NEW]**
> >
> > Dear Reviewer 9Fq5,
> >
> > We would like to remind you about the rebuttal on Image Consistency, as there is not much **time left until the discussion period ends.**
> >
> > Since multiple reviewers have mentioned this issue, **we would like to post an additional rebuttal regarding image consistency.**
> >
> > As shown in the results attached at the anonymous URL below, we demonstrate that **consistency can be improved by utilizing the Runway Gen-3 Alpha Turbo model as a 3D-prior model** instead of Zero123++.
> >
> > **URL:** https://github.com/2024nvs/2024nvs
> >
> > Please kindly note that the core idea of this work is **not the improvement of consistency itself** but rather **the ability to achieve camera control by combining 3D priors with a 3D-free architecture**, which is the novel insight this work has introduced.
> >
> > We hope that **minor artifacts due to our initial choices in implementation using Zero123++ should be alleviated**, after we demonstrate the new results on using Runway Gen-3 Alpha Turbo as 3D prior.
> >
> > We kindly request you to reconsider your rating, given our new implementation and results using Runway Gen-3 Alpha Turbo.
> >
> > With warm regards,
> > The Authors

---

### Official Review · Reviewer_uyxj · 2024-11-02

**Soundness:** 2
**Presentation:** 2
**Contribution:** 3
**Rating:** 8
**Confidence:** 4

**Summary:**

This paper proposes a framework for scene-level camera-control novel view synthesis(NVS). The framework combines 2D image generative prior with object-level 3D generative prior to achieve scene-level NVS. The authors start with an analysis on where should generative model learn the viewpoint information, then proposed a multi-stage inference-time optimization method to generate novel views given specific camera conditions and input images. Qualitative and quantitative evaluation on benchmarks show its superior performance comparing to other methods.

**Strengths:**

1. The idea of combining priors from 2D image generative model and object-level 3D generative model is interesting and can be considered as an effective way to leverage different data source for 3D understanding. I think this idea in general is very important.

2. NVS results on scene images are quite impressive considering no additional training is involved.

**Weaknesses:**

I think this paper's main weakness is on how the authors present the whole framework in a more motivated way. I'd briefly state this weakness here and leave the others in the following questions section.

1. The connection between the section3 and section4 is very confusing. First, the detailed experiment setting in section3 is not clearly explained (see questions), then it suddenly shift from section3 with CLIP embeddings to section4 on leveraging a diffusion model. I suggest authors revise the high-level motivations and present them more clearly in the paper.

2. The figure 2 for pipeline is actually making it harder to understand. I'd suggest at least add some orderings in it can better illustrate the whole pipeline.

**Questions:**

Below are some questions or some parts in the paper that I find confusing.

1. What is the detailed procedure of experiments related to figure 3. and figure 4.? What is the actual model that generates those images? CLIP itself isn't a generative model. Does the author mean using Stable Diffusion with different text embeddings? And how is the image guidance been used in (I assume) this standard SD model?

2. Although inference-time optimization with efficient methods like LoRA and embedding finetuning is not that slow, I'd suggest authors also provide the memory consumption and time used in this process, addition to optimization iterations.

3. Are the specific evaluation viewpoints come from Zero123++ settings? So the method is actually limited to certain viewpoints?

4. It's interesting to see that images generated from Zero123++, which is trained on Objaverse, can generate reasonable images' guidance with scene image input. Combining this with previous question, what's the problem of using some open-sourced scene-level NVS models for pseudo guidance? For example, I'd assume ZeroNVS[1] can provide images with any viewpoint input condition.

[1] Sargent, Kyle, et al. "Zeronvs: Zero-shot 360-degree view synthesis from a single real image." CVPR 2024.

---

> ### Author Response · Authors · 2024-11-22
> **Rebuttal by Authors**
>
> Thank you sincerely for your constructive and detailed feedback. Please find our responses below.
>
> > **(Q1) The connection between the section3 and section4 is very confusing. First, the detailed experiment setting in section3 is not clearly explained (see questions), then it suddenly shift from section3 with CLIP embeddings to section4 on leveraging a diffusion model. I suggest authors revise the high-level motivations and present them more clearly in the paper.**
>
> > **(Q3) What is the detailed procedure of experiments related to figure 3. and figure 4.? What is the actual model that generates those images? CLIP itself isn't a generative model. Does the author mean using Stable Diffusion with different text embeddings? And how is the image guidance been used in (I assume) this standard SD model?**
>
> Above all, we deeply appreciate your constructive feedback on our work.
>
> To elaborate further on Section 3, the experiments were conducted based on a 3D-free stable diffusion-based view synthesis method called HawkI, which utilizes a guidance image. Section 3.1 describes the process of generating images solely from a text prompt, excluding the guidance image required by the base HawkI model, while providing angle-related information.
>
> As you can see, the resulting images are inconsistent, and the camera control angles are not properly followed. Hence, our hypothesis is that the guidance image plays a critical role. This naturally leads to the question of why the guidance image is important.
>
> To explore this, we conducted experiments where $(\alpha_{elev}, \alpha_{azi}) = (30^\circ, 30^\circ)$ was provided, but guidance images were supplied at various angles. The results confirmed that this model heavily relies on the guidance image.
>
> Consequently, our experiments naturally led to the question, **"What role does the guidance image play?"** This inquiry served as the **foundation for deriving our method,** which provides 3D prior information through the guidance image. We hope this conveys the logical progression of our work. We have revised the paper to emphasize the points you mentioned.
>
> We have also clarified the experimental section. As mentioned above, we used HawkI, but we believe there is a possibility of misunderstanding from the reader's perspective that only CLIP itself is being used. Additionally, to eliminate the possibility of misunderstanding that the guidance image is directly incorporated into the Stable Diffusion model (We use a custom pipeline. Stable Diffusion does not follow the approach of providing guidance images), we have made the HawkI reference more explicit. The preprocessed guidance image is passed through a Variational Autoencoder (VAE) to encode it into a latent distribution. This step transforms the image into a compressed latent space representation. So it can provide critical contextual or conditional information to guide the process.
>
> > **(Q2) The figure 2 for pipeline is actually making it harder to understand. I'd suggest at least add some orderings in it can better illustrate the whole pipeline.**
>
> We have updated the method figure based on your suggestions and added step numbers to it. Additionally, we have reorganized the Method section into step-by-step stages to make it easier to understand and improved the paper accordingly.
>
> > **(Q4) Although inference-time optimization with efficient methods like LoRA and embedding finetuning is not that slow, I'd suggest authors also provide the memory consumption and time used in this process, addition to optimization iterations.**
>
> Please refer to A1 of the global rebuttal at the top!
>
> > **(Q5) Are the specific evaluation viewpoints come from Zero123++ settings? So the method is actually limited to certain viewpoints?**
>
> Please refer to A3 of the global rebuttal at the top!

---

> ### Author Response · Authors · 2024-11-22
> **Rebuttal by Authors**
>
> > **(Q6) It's interesting to see that images generated from Zero123++, which is trained on Objaverse, can generate reasonable images' guidance with scene image input. Combining this with previous question, what's the problem of using some open-sourced scene-level NVS models for pseudo guidance? For example, I'd assume ZeroNVS[1] can provide images with any viewpoint input condition. [1] Sargent, Kyle, et al. "Zeronvs: Zero-shot 360-degree view synthesis from a single real image." CVPR 2024.**
>
> Please refer to Global Rebuttal Response A1 for reference.
>
> First, the reason for applying Zero123++ over other models is its **faster image generation speed and minimal resource usage**. For example, ZeroNVS, which you mentioned, allows for a full 360-degree azimuth transformation but cannot handle elevation transformations, which does not align with our goals.
>
> Additionally, when we tested ZeroNVS using an animal image from our HawkI-Real dataset, the results were less natural compared to Zero123++. Furthermore, **generating an image took 7,500 seconds (2 hours and 5 minutes) with significant GPU consumption**, making it even less suitable for our application.
>
> Therefore, we chose to apply the Zero123++ model as it best guarantees the concept of our idea.
>
> **Given the improved results/methodology presented in this revision, and your initial assessment of Acceptance (6), we would be deeply grateful if you could raise your review ratings that better reflect your encouraging evaluation!**
>
> Thank you sincerely for taking the time to provide such detailed reviews. If you have any additional questions, please feel free to reach out at any time!

---

> > ### Comment · Area_Chair_5oV2 · 2024-11-24
> > **Discussion Period Ending Soon**
> >
> > Dear Reviewer,
> >
> > The discussion period will end soon. Please take a look at the author's comments and begin a discussion.
> >
> > Thanks, Your AC

---

> ### Author Response · Authors · 2024-11-25
> **Official Comment by Authors**
>
> Dear Reviewer,
>
> We would like to sincerely thank you once again for the time and effort you have devoted to reviewing our paper. With the discussion period ending in two days, we would greatly appreciate it if you could review our rebuttal at your convenience, should your schedule permit!
>
> If you have any questions, require additional explanations, or see any areas that need improvement, please feel free to let us know, and we will address them immediately!
>
> We are truly grateful for the valuable feedback you have provided on our research. Additionally, considering the improved methodology and results in the current version, we would be sincerely thankful if you could consider raising your review rating!!!
>
> Hope you have a vibrant start to the week! Thank you very much!!!
>
> With warm regards,
> The Authors

---

> > ### Comment · Reviewer_uyxj · 2024-11-25
> > **Official Comment from Reivewer uyxj**
> >
> > I checked the author's response and other reviews.
> >
> > I think the response answers my previous questions, and I appreciate the authors' efforts in revising main paper in better illustrating the motivations. I understand the time may be very short in this discussion period, so if upon acceptance I hope authors can still revise the main figure to better convey the whole pipeline. Other parts look good to me.
> >
> > I agree with the positive contribution of this paper so I'll maintain my positive ratings. Thanks again for the authors' response.

---

> ### Author Response · Authors · 2024-11-25
>
> Dear Reviewer uyxj,
>
> First, I would like to sincerely thank you for maintaining a positive review rating. Additionally, thank you for highlighting the importance of our initial paper's idea and the impressive results as its strengths.
>
> Regarding the mentioned Figure 2, As you mentioned, **we have revised the numbering to align with the numbering in the method section.** We will further consider ways to make it more impactful for readers and improve it accordingly. We will also do our utmost to address the points you raised, even after acceptance.
>
> I am reaching out to kindly ask if you might reconsider raising the review rating, if you mind!
>
> There have been further developments in the idea you highlighted as a strength, the results have been further refined, and we have thoroughly addressed and revised all the points you mentioned, with most of the highlighted areas improved.
>
> Given such improvement, positive feedback, and your final 'Accept' evaluation, **I would respectfully suggest considering a higher score that better reflects your encouraging assessment!!!**
>
> Thank you very much!!!
>
> With warm regards,
> The Authors

---

### Official Review · Reviewer_BRTu · 2024-11-03

**Soundness:** 2
**Presentation:** 3
**Contribution:** 2
**Rating:** 5
**Confidence:** 4

**Summary:**

The paper addresses the problem of generating novel views from a single image without relying on 3D datasets for training. The authors propose to leverage the 3D knowledge embedded in large-scale pretrained diffusion models to create camera-controlled viewpoints through test-time optimization. Specifically, the method fine-tunes the text prompt and diffusion model to align with the input and a pseudo-guidance image, incorporating mutual information guidance during the final diffusion process to enhance image fidelity. The experiments demonstrate that the proposed model can generate semantically consistent novel views for images with complex background.

**Strengths:**

- The proposed approach effectively integrate 3D-free methods and pretrained 3D-based prior to achieve viewpoint-controlled novel-view synthesis. This technique can generalize to complex scenes without needing large 3D training datasets.

- The mutual information guidance improves fidelity of the generated images over the pseudo guidance.

- The method is tested both qualitatively and quantitatively, showcasing superior performance compared to state-of-the-art models, especially in handling background details and complex scenes.

- The analysis of the significance of CLIP and guidance images provides a solid motivation for the proposed research.

**Weaknesses:**

- While the authors acknowledge the limitation of inference-time optimization in terms of real-time applicability and scalability, a direct runtime comparison with baseline models would provide additional clarity.

- This method is incapable of generating arbitrary camera viewpoints. this might be due to the choice of the 3D prior, Zero123++, which can provide guidance images at six fixed viewpoints. Clarifying this limitation or demonstrating the capability of generating arbitrary viewpoints is much needed.

- The novelty of the proposed framework is somewhat limited. The approach builds heavily on the HawkI, with modifications such as replacing guidance images generated via Inverse Perspective Mapping with those from a 3D prior method (Zero123++). This modification, while practical, could be seen as a straightforward combination.

**Questions:**

The generated images across different viewpoints appear inconsistent. Does the proposed method include any mechanisms to encourage consistency across these viewpoints?

---

> ### Author Response · Authors · 2024-11-22
> **Rebuttal by Authors**
>
> Thank you sincerely for your constructive and detailed feedback. Please find our responses below.
>
> > **(Q1) While the authors acknowledge the limitation of inference-time optimization in terms of real-time applicability and scalability, a direct runtime comparison with baseline models would provide additional clarity.**
>
> Please refer to A1 of the global rebuttal at the top!
>
> > **(Q2) This method is incapable of generating arbitrary camera viewpoints. this might be due to the choice of the 3D prior, Zero123++, which can provide guidance images at six fixed viewpoints. Clarifying this limitation or demonstrating the capability of generating arbitrary viewpoints is much needed.**
>
> Please refer to A3 of the global rebuttal at the top!
>
> > **(Q3) The novelty of the proposed framework is somewhat limited. The approach builds heavily on the HawkI, with modifications such as replacing guidance images generated via Inverse Perspective Mapping with those from a 3D prior method (Zero123++). This modification, while practical, could be seen as a straightforward combination.**
>
> Please refer to A2 of the global rebuttal at the top!
>
> > **(Q4) The generated images across different viewpoints appear inconsistent. Does the proposed method include any mechanisms to encourage consistency across these viewpoints?**
>
> Please refer to A2 of the global rebuttal at the top!
>
> **Considering the improved results/methodology presented in this revision, and considering your positive evaluation regarding approach/method/result/analysis, we would be deeply grateful if you could raise your review ratings based on these enhancements.**
>
> Thank you sincerely for taking the time to provide such detailed reviews. If you have any additional questions, please feel free to reach out at any time!

---

> > ### Comment · Area_Chair_5oV2 · 2024-11-24
> > **Discussion Period Ending Soon**
> >
> > Dear Reviewer,
> >
> > The discussion period will end soon. Please take a look at the author's comments and begin a discussion.
> >
> > Thanks, Your AC

---

> ### Author Response · Authors · 2024-11-25
>
> Dear Reviewer,
>
> We would like to sincerely thank you once again for the time and effort you have devoted to reviewing our paper. With the discussion period ending in two days, we would greatly appreciate it if you could review our rebuttal at your convenience, should your schedule permit!
>
> If you have any questions, require additional explanations, or see any areas that need improvement, please feel free to let us know, and we will address them immediately!
>
> We are truly grateful for the valuable feedback you have provided on our research. Additionally, considering the improved methodology and results in the current version, we would be sincerely thankful if you could consider raising your review rating!!!
>
> Hope you have a vibrant start to the week! Thank you very much!!!
>
> With warm regards,
> The Authors

---

> > ### Comment · Reviewer_BRTu · 2024-11-25
> >
> > The rebuttal adequately addressed my concerns regarding runtime comparison. However, I still have reservations about the issue of image consistency. One crucial criterion for novel view synthesis from any source views is generating consistent images across viewpoints. Unfortunately, the rebuttal did not provide sufficient quantitative evaluation on this aspect, and the qualitative results presented were not entirely convincing. This also raises further concerns about the novelty of the approach. The authors argue that the regularization term, which is the main novelty of this paper, is critical to novel view synthesis. If the results were impressive, then the regularization term would be a simple but effective solution, which would be a great insight. However, the current results do not fully support the authors' claims. Hence, my doubt on novelty remains.

---

> ### Author Response · Authors · 2024-11-25
> **Official Comment by Authors (1)**
>
> Dear Reviewer BRTu,
>
> First of all, we would like to thank you for pointing out this important aspect. While the rebuttal itself does not include experimental results, we would like to highlight that **all the results have been updated in the paper.** Specifically, we have conducted quantitative and qualitative evaluations on the inclusion and exclusion of the regularization loss, as emphasized in Section 5.3, Figure 7, and Table 2.
>
> In particular, the examples highlighted in the ablation study shown in Figure 7 include scenes of a Hawaiian beach and a waterfall. In both cases, the Zero123++ model failed to provide accurate viewpoints and guidance to the model. However, the application of the regularization loss **significantly improved the consistency of the textures in the Hawaiian beach scene and enhanced the consistency of the rock textures generated in the waterfall scene.**
>
> We kindly ask you to refer to Section 5.3 for more detailed information on this aspect.
>
> > *Part of sentences from Section 5.3*: "For the Hawaiian beach scene, the **output generated with the regularization term exhibits a more consistent style** compared to the output produced without it. Despite the inaccuracies in the Zero123++ guidance images, the regularization term facilitates **more reliable camera control than the results generated without it.**
> Similarly, in the waterfall scene, the regularization term enhances the consistency of the generated rock textures surrounding the waterfall. **Without the regularization term, these textures are inconsistently represented; however, with it, the style is maintained more faithfully.** Once again, Zero123++ does not provide accurate guidance in this case, underscoring the significant contribution of the regularization loss to improved control and visual coherence in the generated images."
>
> We also conducted an **ablation study** to compare the results with and without the application of the regularization loss. By analyzing seven image evaluation metrics presented in the existing evaluation — LPIPS, CLIP, DINO, SSCD, CLIP-I, PSNR, and SSIM — we demonstrated superior performance, thereby providing quantitative evaluations. We have attached the results in a table below for your reference, and we would appreciate it if you could take a look.
>
> > | Dataset, Angle, Method                       | LPIPS ↓    | CLIP ↑    | DINO ↑    | SSCD ↑    | CLIP-I ↑   | PSNR ↑    | SSIM ↑    |
> |----------------------------------------------|------------|-----------|-----------|-----------|------------|-----------|-----------|
> | **HawkI-Syn (30°, 30°) w/ regularization**   | **0.5661** | **29.9563** | **0.4314** | 0.3638    | **0.8317** | **11.0664** | **0.3162** |
> | HawkI-Syn (30°, 30°) w/o regularization      | 0.5867     | 28.5417    | 0.4122    | **0.3640** | 0.8243     | 10.8272    | 0.2954     |
> | **HawkI-Real (30°, 30°) w/ regularization**  | **0.6201** | **29.8850** | 0.3346    | **0.2588** | 0.8152     | **9.4009** | **0.2184** |
> | HawkI-Real (30°, 30°) w/o regularization     | 0.6257     | 29.0798    | **0.3357** | 0.2401    | **0.8231** | 9.1957     | 0.2014     |
> | **HawkI-Syn (30°, 270°) w/ regularization**  | **0.5744** | **29.1800** | **0.4148** | **0.3684** | **0.8327** | **11.0661** | **0.3047** |
> | HawkI-Syn (30°, 270°) w/o regularization     | 0.5952     | 28.9866    | 0.4098    | 0.3350    | 0.8248     | 10.8656    | 0.2850     |
> | **HawkI-Real (30°, 270°) w/ regularization** | **0.5868** | **30.5489** | **0.4126** | **0.3424** | **0.8708** | **10.6177** | **0.2687** |
> | HawkI-Real (30°, 270°) w/o regularization    | 0.6114     | 29.9184    | 0.4003    | 0.3075    | 0.8541     | 10.2958    | 0.2615     |
> | **HawkI-Syn (-20°, 210°) w/ regularization** | **0.5740** | **29.1144** | **0.4277** | 0.3529    | **0.8280** | **10.9697** | **0.2837** |
> | HawkI-Syn (-20°, 210°) w/o regularization    | 0.5860     | 28.6385    | 0.3969    | **0.3559** | 0.8171     | 10.8401    | 0.2792     |
> | **HawkI-Real (-20°, 210°) w/ regularization**| **0.6185** | **30.6729** | 0.3610    | **0.2880** | **0.8448** | **10.3130** | **0.2223** |
> | HawkI-Real (-20°, 210°) w/o regularization   | 0.6338     | 29.1693    | **0.3817** | 0.2605    | 0.8263     | 10.0794    | 0.2117     |
> | **HawkI-Syn (-20°, 330°) w/ regularization** | **0.5624** | **29.2144** | **0.4487** | **0.3892** | **0.8559** | **11.2175** | **0.3048** |
> | HawkI-Syn (-20°, 330°) w/o regularization    | 0.5714     | 28.4089    | 0.4476    | 0.3870    | 0.8492     | 11.0409    | 0.2947     |
> | **HawkI-Real (-20°, 330°) w/ regularization**| 0.5925     | **29.5090** | **0.3899** | **0.3127** | **0.8689** | **10.6183** | **0.2971** |
> | HawkI-Real (-20°, 330°) w/o regularization   | **0.5894** | 28.8531    | 0.3704    | 0.2954    | 0.8506     | 10.5213    | 0.2828     |
>
> > Table 2: Quantitative Results of Ablation Study. Evaluation of seven metrics demonstrates the superior results of the regularized method over the non-regularized one.

---

> ### Author Response · Authors · 2024-11-25
> **Official Comment by Authors (2)**
>
> The quantitative and qualitative evaluations you mentioned demonstrate that applying the **regularization loss term is an effective solution for maintaining consistency.**
>
> Additionally, as **highlighted in the initial Strengths section that you wrote,** we kindly ask you to consider the fact that **this model is fundamentally a 3D-free method and operates under the constraint of no additional training.**
>
> Most novel view synthesis (that achieves the 'crucial criterion' you mentioned) papers **rely on training with 3D datasets**, whereas **our approach does not.**
>
> It is within these constraints that we introduced the regularization loss term to address viewpoint consistency. We would greatly appreciate it if you could reconsider your evaluation, taking into account the strong results achieved **without training using 3D datasets.**
>
> With warm regards,
> The Authors

---

> > ### Author Response · Authors · 2024-11-28
> >
> > Dear Reviewer BRTu,
> >
> > Could you confirm **if you’ve reviewed our response and whether it or the subsequent discussions have influenced your evaluation?**
> >
> > Thank you for taking the time to review our work; your insights have been incredibly helpful to our research. It is **truly meaningful for us to have someone like you review our paper.**
> >
> > We sincerely appreciate your encouragement of our research potential, **and we would be deeply grateful if you could consider raising the review rating.**
> >
> > Sincerely,
> > The Authors

---

> > > ### Comment · Reviewer_BRTu · 2024-11-28
> > >
> > > Thank you for the clarification. While the proposed method quantitatively outperforms the baseline, I still have reservations about the significance of this improvement. The inconsistency across different viewpoints remains noticeable. Papers like Zero123++ have demonstrated consistency by reconstructing complete 3D objects/scenes and evaluating the rendering quality. Unfortunately, the authors have not provided similarly strong evidence to support their claims in this paper. Given the concern above, I will maintain my original score.

---

> > > > ### Author Response · Authors · 2024-11-28
> > > >
> > > > Dear Reviewer BRTu,
> > > >
> > > > Thank you for pointing out this important issue. Since you raised the concern about inconsistency, I would like to provide additional clarification on this matter.
> > > >
> > > > As I mentioned, models like Zero123++ are trained **using 3D datasets.** However, **our model is not.**
> > > >
> > > > Therefore, we kindly ask that the results of our model be evaluated from a **different perspective compared to NVS models trained on conventional 3D datasets.**
> > > >
> > > > Below, I have included the key points from our previous rebuttal for your reference.
> > > >
> > > > > Additionally, as highlighted in the initial Strengths section that you wrote, we kindly ask you to consider the fact that this model is **fundamentally a 3D-free method and operates under the constraint of no additional training.**
> > > >
> > > > > Most novel view synthesis (that achieves the 'crucial criterion' you mentioned) papers **rely on training with 3D datasets, whereas our approach does not.**
> > > >
> > > > > It is within these constraints that we introduced the regularization loss term to address viewpoint consistency. We would greatly appreciate it if you could reconsider your evaluation, taking into account the **strong results (also you mentioned in the initial review) achieved without training using 3D datasets.**
> > > >
> > > > Despite these constraints, the consistency of Figures 5 and 6 has been maintained.
> > > >
> > > > Specifically, in the case of the pyramid, unlike the existing Zero123++, HawkI, and StableZero123 models, the position of the dark side that should be reflected at a specific angle was accurately captured. For the waterfall, the position of the water stream; for the house image, details such as the chimney and the number of windows for each image were accurately reflected.
> > > >
> > > > As for Figure 6, as explained in the main text, the position of the Seine River in the Eiffel Tower scene and the Hawaii beach described above were accurately reflected. Even in the case of the cat photo, the model preserved its form without distortion, accurately reflecting the fur color and the position of the cat’s legs. Again, please keep in mind that all the results were generated **despite the poor quality of the input images for the 3D-prior model, Zero123++, and evaluate the outcomes accordingly.**
> > > >
> > > > In your initial review, the following content was mentioned under 'Strengths': 'The method is tested both qualitatively and quantitatively, showcasing **superior performance** compared to state-of-the-art models, especially in **handling background details and complex scenes.**'
> > > >
> > > > Additionally, we have provided a thorough explanation regarding the question you raised: 'Does the proposed method include **any mechanisms to encourage consistency across these viewpoints?**'
> > > >
> > > > **Based on your feedback, we incorporated algorithms** addressing 'encourage consistency across these viewpoints' **into the initial paper.** If our paper is accepted, we will do our utmost to address the points you mentioned and submit a thoroughly revised camera-ready paper.
> > > >
> > > > As a result, the Reviewer (uyxj), who initially rated the paper a 6, **raised their Review Rating to 8.** Reviewer hAuB also stated that the evaluation of this paper **should be increased from 6 to 7.** Considering these points collectively, we would like to respectfully ask if you could reconsider your Review Rating.
> > > >
> > > > We fully respect your opinion and understand that everyone views results from different perspectives. However, **since your review also acknowledges the merits** of this work, and the authors have **diligently addressed your concerns**, and because the **majority of reviewers have expressed support for this work**, we would sincerely appreciate it if you could provide encouragement for our work through your rating.
> > > >
> > > > With warm regards,
> > > > The Authors

---

> ### Author Response · Authors · 2024-11-30
>
> Dear Reviewer BRTu,
>
> We greatly appreciate your time and effort in reviewing our work and providing detailed feedback. Your insights have been invaluable in helping us refine our paper. As the discussion period is coming to a close in a few days, we kindly request you review our third-round rebuttal at your earliest convenience.
>
> Thank you once again for your thoughtful consideration.
>
> With warm regards,
> The Authors

---

> > ### Author Response · Authors · 2024-12-02
> > **Author Rebuttal - Image Consistency [NEW]**
> >
> > Dear Reviewer BRTu,
> >
> > Since multiple reviewers have mentioned this issue, **we would like to post an additional rebuttal regarding image consistency.**
> >
> > As shown in the results attached at the anonymous URL below, we demonstrate that **consistency can be improved by utilizing the Runway Gen-3 Alpha Turbo model as a 3D-prior model** instead of Zero123++.
> >
> > **URL:** https://github.com/2024nvs/2024nvs
> >
> > Please kindly note that the core idea of this work is **not the improvement of consistency itself** but rather **the ability to achieve camera control by combining 3D priors with a 3D-free architecture**, which is the novel insight this work has introduced.
> >
> > We hope that **minor artifacts due to our initial choices in implementation using Zero123++ should be alleviated**, after we demonstrate the new results on using Runway Gen-3 Alpha Turbo as 3D prior.
> >
> > We kindly request you to reconsider your rating, given our new implementation and results using Runway Gen-3 Alpha Turbo.
> >
> > With warm regards,
> > The Authors

---

> > > ### Author Response · Authors · 2024-12-03
> > > **Author Rebuttal Reminder - Image Consistency [NEW]**
> > >
> > > Dear Reviewer BRTu,
> > >
> > > We would like to remind you about the rebuttal on Image Consistency, as there is not much **time left until the discussion period ends.**
> > >
> > > Since multiple reviewers have mentioned this issue, **we would like to post an additional rebuttal regarding image consistency.**
> > >
> > > As shown in the results attached at the anonymous URL below, we demonstrate that **consistency can be improved by utilizing the Runway Gen-3 Alpha Turbo model as a 3D-prior model** instead of Zero123++.
> > >
> > > **URL:** https://github.com/2024nvs/2024nvs
> > >
> > > Please kindly note that the core idea of this work is **not the improvement of consistency itself** but rather **the ability to achieve camera control by combining 3D priors with a 3D-free architecture**, which is the novel insight this work has introduced.
> > >
> > > We hope that **minor artifacts due to our initial choices in implementation using Zero123++ should be alleviated**, after we demonstrate the new results on using Runway Gen-3 Alpha Turbo as 3D prior.
> > >
> > > We kindly request you to reconsider your rating, given our new implementation and results using Runway Gen-3 Alpha Turbo.
> > >
> > > With warm regards,
> > > The Authors

---

### Author Response · Authors · 2024-11-22
**Author Rebuttal - Global Response (1)**

First of all, we would like to sincerely thank you for providing detailed comments on our research and offering constructive feedback. Among the comments mentioned by various reviewers, overlapping points have been summarized and addressed below.

- **We would appreciate it if you could read it along with the revised paper we have updated.**
- **Also, we kindly request that you review the paper again as there have been improvements in the results/methodology.**

### **(A1) Computation Time and GPU Memory Consumption (Reviewers BRTu, uyxj, 9Fq5, hAuB)**

Many reviewers pointed out the need for an analysis of computation time, so we conducted related experiments. By comparing the computation time and memory consumption of the Zero123++, Stable Zero123, and ZeroNVS models, we quantitatively demonstrated which model is best suited for our framework.

Testing on an A100 GPU (40GB) showed that **Zero123++ (10.18GB)** consumes the **least memory** compared to Stable Zero123 (39.3GB) and ZeroNVS (33.48GB). Additionally, the computation time for generating a single camera-controlled image was 1,278 seconds for Stable Zero123 and 7,500 seconds for ZeroNVS, while **Zero123++ could generate an image in just 20 seconds**.

As a result, Zero123++ was selected as the guidance model for our framework.

> | Model           | Memory Consumption      | Computation Time |
|------------------|-------------------------|------------------|
| Zero123++       | **10.18 GB** / 40.0 GB (9,715 MiB)  | **20 sec**          |
| Stable Zero123  | 39.3 GB / 40.0 GB (37,479 MiB)  | 1,278 sec       |
| ZeroNVS         | 33.48 GB / 40.0 GB (31,929 MiB) | 7,500 sec       |

> Table 1. **Comparison of computation times for 3D-prior models.** Among the prior works in NVS frequently mentioned, including Zero123++, Stable Zero123, and ZeroNVS, the Zero123++ model has the shortest computation time. Our research applies the Zero123++ model, which has the lowest computation time among 3D-prior models, to obtain 3D prior information without requiring any additional computation time.

We understand concerns about the potential increase in computation time due to the addition of 3D-prior framework.

However, experimental results showed that the computation time of our model (measured as the time from program initiation to generating a first image) was **393 seconds**, which is slightly faster than the 401 seconds of the baseline model.

Throughout the optimization/inference process, **our model uses at most 8.43GB** of memory, and the Zero123++ image generation process requires 10.18GB, which indicates that our approach is viable even on consumer-grade GPUs.

This **highlights the value of our method in terms of general applicability.** The computation time and memory consumption analysis can be found in the revised paper.

> | Model            | Step           | Memory Consumption         | Computation Time          |
|-------------------|----------------|-----------------------------|---------------------------|
| HawkI            | Optimization   | 7.20 GB / 40.0 GB (6,875 MiB) | 395 sec                  |
|                   | Inference      | 8.43 GB / 40.0 GB (8,045 MiB) | 6 sec (each image)       |
| Total                  |       | **8.43 GB** / 40.0 GB (8,045 MiB) | **401 sec**       |
| Ours (w/o regloss) | Zero123++     | 10.18 GB / 40.0 GB (9,715 MiB) | 20 sec                  |
|                   | Optimization   | 7.21 GB / 40.0 GB (6,879 MiB) | 372 sec                 |
|                   | Inference      | 8.43 GB / 40.0 GB (8,049 MiB) | 6 sec (each image)       |
| Total                  |       | **10.18 GB** / 40.0 GB (9,715 MiB) | **398 sec**       |
| Ours (w/ regloss) | Zero123++      | 10.18 GB / 40.0 GB (9,715 MiB) | 20 sec                  |
|                   | Optimization   | 7.21 GB / 40.0 GB (6,885 MiB) | 367 sec                 |
|                   | Inference      | 8.43 GB / 40.0 GB (8,045 MiB) | 6 sec (each image)       |
| Total                  |       | **10.18 GB** / 40.0 GB (9,715 MiB) | **393 sec**       |

> Table 2. **Detailed Step-wise Comparison**. Even when applying Zero123++ to our methodology, the additional GPU memory consumption is relatively small, at **2.97GB** (10.18GB - 7.21GB), and it takes only **20 seconds** to generate the guidance image using Zero123++. From an overall perspective, HawkI takes 401 seconds to complete optimization and generate the first image through inference, while Ours (w/o regloss) takes 398 seconds, and Ours (w/ regloss) takes **393 seconds**. This demonstrates that our methodology does not result in significant differences in computation time or memory consumption while achieving better performance compared to existing methods. Total memory consumption refers to the worst case, the computation time indicates the total execution time. *i.e.,* the time taken for the model to run and output the first image.

---

> ### Author Response · Authors · 2024-11-22
> **Author Rebuttal - Global Response (2)**
>
> ### **(A2) Results and Idea Improvement Through Regularization Loss (Reviewers BRTu, 9Fq5, hAuB, yRXV)**
>
> Given that the CLIP model lacks an understanding of camera control information, it is essential to leverage 3D prior information from pretrained model to address this limitation.
>
> By incorporating this prior knowledge, our goal is to enhance the CLIP model's understanding of viewpoints (e.g., elevation and azimuth) to generate the desired camera control outputs.
>
> **To address issues such as maintaining consistency across angles in the same image and improving image generation performance in complex scenes**, as pointed out by the reviewers, we introduced a regularization term between the text embedding containing elevation and azimuth information ($e_{target}$) and the optimized text embedding ($e_{view}$) in addition to the pretrained guidance model.
>
> This regularization term **not only enriches viewpoint knowledge in 3D space but also addresses the limitations of the 3D-prior guidance model**, which struggles with complex scenes.
>
> By applying this regularization loss, we hypothesize that we can improve camera control results by constructing a model that uses the guidance image as supplementary information rather than relying solely on it. To improve viewpoint consistency, we added this additional regularization loss term to the reconstruction loss. The term, calculated as
>
> $L_{\text{reg}} = \| e_{\text{view}} - e_{\text{target}} \|^2$
>
> provides a constraint between the embedding $e_{\text{view}}$ and the intended angular information $e_{\text{target}}$, reinforcing the target viewpoint in the generated outputs.
>
> Consequently, images predicted using the 3D-based NVS method, Zero123++, serve as weak or pseudo guidance. The optimization strategy conditions the embedding space with knowledge related to the input image and its view variants, leveraging the guidance image's prior knowledge to facilitate view transformation and guide viewpoint adjustments.
>
> **Through this approach, we have addressed the concerns raised about image consistency and camera control performance in complex scenes.** The results and detailed implementation details can be found in the **revised paper**.
>
> ### **(A3) Limitations of Zero123++ (Reviewers BRTu, uyxj, 9Fq5)**
>
> Thank you for your important comments regarding the limitations of Zero123++.
>
> While Zero123++ has the drawback of being unable to generate images from specific angles, **it was chosen due to its speed and minimal impact on computation time.** For instance, although ZeroNVS (Thanks to reviewer uyxj) can perform azimuth transformations (i.e., 360-degree rotation), it cannot handle elevation transformations, which are essential for our task. In addition, ZeroNVS requires 7,500 seconds to generate an image, making it unsuitable for general use. Moreover, Stable Zero123 has a memory consumption of 39.3GB and takes 1,278 seconds to generate an image.
>
> **Therefore, applying Zero123++ is the best choice to emphasize the concept of our framework, given its speed and practical utility.** Zero123++ also consumes approximately 10.18GB of memory, making it feasible to run even on consumer-grade GPUs, thus enhancing its applicability. This was briefly mentioned in the previous version's qualitative analysis, and in this revision, we have explicitly stated in the conclusion that addressing these limitations will be a focus of future research.
>
> ### **Closing Remarks**
>
> We sincerely thank **reviewers BRTu, uyxj, and hAuB for positively evaluating the potential and results** of our research.
>
> **Considering the improved results/methodology presented in this revision, we would be deeply grateful if you could reassess your review ratings based on these enhancements.**
>
> Once again, we sincerely thank you for your constructive feedback and for recognizing the value of our ideas.
>
> We will provide individual responses to any additional comments in separate replies.

---

### Author Response · Authors · 2024-12-02
**Author Rebuttal - Global Response : Image Consistency [NEW]**

Dear Reviewers,

Since multiple reviewers have mentioned this issue, **we would like to post an additional rebuttal regarding image consistency.**

As shown in the results attached at the anonymous URL below, we demonstrate that **consistency can be improved by utilizing the Runway Gen-3 Alpha Turbo model as a 3D-prior model** instead of Zero123++.

**URL:** https://github.com/2024nvs/2024nvs

Please kindly note that the core idea of this work is **not the improvement of consistency itself** but rather **the ability to achieve camera control by combining 3D priors with a 3D-free architecture**, which is the novel insight this work has introduced.

We hope that **minor artifacts due to our initial choices in implementation using Zero123++ should be alleviated**, after we demonstrate the new results on using Runway Gen-3 Alpha Turbo as 3D prior.

We kindly request **Reviewer BRTu, 9Fq5, and yRXV to reconsider their ratings**, given our new implementation and results using Runway Gen-3 Alpha Turbo.

With warm regards,
The Authors

---

### Meta-Review · Area_Chair_5oV2 · 2024-12-21

**Metareview:**

The paper addresses the problem of scene-level novel view synthesis (given an image and a new camera position, generate a new image). The paper proposes to combine a "3d-free" method which uses CLIP and a image guidance with a "3d prior", an existing model which has already been trained to do novel view synthesis (though on objects). The paper demonstrates strong results on the HawkI dataset.

The strength of the paper is its quantitative performance on HawkI dataset. The weakness are numerous:
1) The evaluation on this HawkI dataset for scene-level 3d is really abnormal. It avoids many more standard datasets such as skipping over many more standard dataset such as CO3D, RE10K, Acid or DTU.
2) The comparisons used in the papers are weak and the rebuttal is literally wrong. As many reviewers mentioned, ZeroNVS should be a standard comparison. In the rebuttal the authors state that "generating a single camera-controlled image was 7,500 seconds for ZeroNVS". This is inaccurate. ZeroNVS follows the architecture of Zero123 which takes 2 seconds on a RTX A6000 GPU. It's possible that the NeRF optimization was included but this is not necessary for a single image sample and other papers, e.g. MegaScenes have disentangled the diffusion inference from the NeRF optimization in order to compare directly with the NVS properties.
3) The writing and method are confusing. The paper repeatedly says that their method does not use 3D data. But they directly use Zero123+ which is trained on 3D data. In the rebuttal, they clarify and say that they don't use "additional" 3D data and that Zero123++ is object-level. While this is true, it runs completely against how the current intro is written. Even a one of the more positive reviewers (hAuB) notes this in their weakness (Weakness 4).
4) Even the qualitative performance on HawkI seems to lack image consistency.

I suggest rejection. For future submissions, the authors should use standard datasets and compare against the SOTA or near SOTA for these datasets (e.g. MegaScenes would be an approximate example).

**Additional Comments On Reviewer Discussion:**

Reviewers yRXV and 9Fq5 both rate the paper a 3, rejection with the main concerns being the inconsistencies in the generated images and the comparisons.

Reviewer BRTu rates the paper a 5 weak reject raising concerns about the novelty and comparisons.

Reviewer hAuB rates the paper a 6 saying that the paper has weaknesses but offers something to the community.

Reviewer uyxj rates the paper a 8 saying that the NVS results on scene images are "impressive considering no additional training is involved" and saying that this idea of combining 2D image and object-level 3D is important. I respectfully disagree. Considerable effort is being made to acquire scene-level pose data (both synthetic and real). Many papers just train on this data now.

In general, I think the lack of comparisons on standard datasets and with standard methods (like ZeroNVS) mean that this paper should be by default rejected. I also think the misleading writing and limited novelty hurt this paper.

---

### Decision · Program_Chairs · 2025-01-22

Reject